

# Development and prospects of the regional MiKlip decadal prediction system over Europe: Predictive skill, added value of regionalization and ensemble size dependency

Mark Reyers[1], Hendrik Feldmann[2], Sebastian Mieruch[2,3], Joaquim G. Pinto[2], Marianne Uhlig[2,4], Bodo Ahrens[5], Barbara Früh[6], Kameswarrao Modali[7], Natalie Laube[2], Julia Mömken[1,2], Wolfgang Müller[7], Gerd Schädler[2], Christoph Kottmeier[2]

[1]Institute for Geophysics and Meteorology, University of Cologne, Cologne, Germany
[2]Institute for Meteorology and Climate Research (IMK-TRO), Karlsruhe Institute of Technology (KIT), Karlsruhe, Germany
[3]Alfred-Wegener Institute for Polar and Marine Sciences, Bremerhaven, Germany
[4]School of Geography, Environment and Earth Sciences, Victoria University of Wellington, Wellington, New Zealand
[5]Institute for Atmospheric and Environmental Sciences, Goethe-University Frankfurt a.M., Frankfurt a.M., Germany
[6]Deutscher Wetterdienst (DWD), Offenbach, Germany
[7]Max Planck Institute for Meteorology, Hamburg, Germany

Correspondence to: M. Reyers, (mreyers@meteo.uni-koeln.de)

**Abstract.** The current state of development and prospects of the regional MiKlip decadal prediction system for Europe are analysed. The Miklip regional system consists of two 10-member hindcast ensembles computed with the global coupled model MPI-ESM-LR downscaled for the European region with COSMO-CLM to a horizontal resolution of 0.22° (~25km). Prediction skills are computed for temperature, precipitation, and wind speed using E-OBS and an ERA-Interim driven COSMO-CLM simulation as verification datasets. Focus is given to the eight European PRUDENCE regions and to lead years 1-5 after initialization. Evidence of the general potential for regional decadal predictability for all three variables is provided. For example, the initialized hindcasts outperform the uninitialized historical runs for some key regions in Europe and for some variables both in terms of accuracy and reliability. However, forecast skill is not detected in all cases, but it depends on the variable, the region, and the hindcast generation. A comparison of the downscaled hindcasts with the global MPI-ESM-LR runs reveals that the MiKlip prediction system may distinctly benefit from regionalization, in particular for parts of Southern Europe and for Scandinavia. The forecast accuracy and the reliability of the MiKlip ensemble is systematically enhanced when the ensemble size is stepwise increased, and a number of 10 members is found to be suitable for decadal predictions. This result is valid for all variables and European regions in both the global and regional MiKlip ensemble. The predictive skill improves distinctly, particularly for temperature, when retaining the long-term trend in the time series. The present results are encouraging towards the development of a regional decadal prediction system.



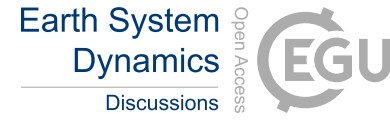

# 1. Introduction

In recent years, the interest in climate predictions on time-scales from one year up to a decade has increased in the climate science community, since this time span falls within the planning horizon for a wide variety of decision makers (Meehl et al., 2009; 2014). A large ensemble of initialised decadal hindcasts has been consolidated in a component of the Coupled Model Intercomparison Project Phase 5 (CMIP5; Taylor et al., 2012), and the number of studies aiming at decadal predictions has strongly increased in recent years (for a review see Meehl et al., 2014). Typically, the North Atlantic is a key region for decadal predictions and forecast skill is found for various quantities such as heat content and SST (e.g. Kröger et al, 2012; Yaeger et al., 2012), CO2 uptake (Li et al., 2016) and integrated quantities such as the AMOC (Pohlmann et al., 2013a) and sub-polar gyre (Matei et al., 2012; Yaeger et al., 2012; Robson et al., 2013). Other studies focus on primary meteorological parameters on the global scale, in particular surface temperature (e.g Chikamoto et al., 2012; Doblas-Reyes et al., 2013; Ho et al., 2013; Corti et al., 2015), while few studies analyse storm tracks (Kruschke et al., 2014, 2016), Atlantic tropical cyclones (Dunestone et al., 2011), intense or extreme events (e.g. Benestad and Mezghani, 2015) or zoom into a certain region of the world (e.g. Guemas et al., 2015). For example, Sutton and Hodson (2005) found a downstream impact of the Atlantic Multidecadal Oscillation (AMO; SST anomalies over the North Atlantic) on decadal time scales, with higher temperatures and increased precipitation over Europe in an AMO warm phase compared to a cold phase.

In the German research consortium MiKlip (http://www.fona-miklip.de), a global decadal prediction system was developed based on the Max-Planck-Institute Earth System Model (MPI-ESM) (for an overview see Marotzke et al., 2016). Within the first phase of the project, three hindcast generations were produced. The skill of the MiKlip System for decadal predictions was analysed in a wide variety of recent studies. For example, Müller et al. (2012) investigated global surface air temperature in the first generation of the global MiKlip system (baseline0) and found that the initialized hindcasts have predictive skill over the North-Atlantic region, while negative skill scores are identified for the tropics. A modified initialization in the second global MiKlip system generation (baseline1) considerably improves the performance in the tropics, but brings only limited skill improvement over the North-Atlantic and Europe (Pohlmann et al., 2013b). Significant positive skill scores for cyclone frequencies over the Central North-Atlantic were identified by Kruschke et al. (2014) in the global baseline0 and baseline1 generations, but no significant skill was detected over the Eastern North-Atlantic and Europe. Furthermore, Kadow et al. (2016) evaluated the global MiKlip system with respect to temperature and precipitation, giving evidence that an enlargement of the hindcast ensemble generally leads to an improvement of the prediction system.

The MiKlip consortium is to our best knowledge the first institution worldwide which has established a decadal prediction system for the regional scale. With this aim considerable efforts were made to downscale the global MPI-ESM hindcasts by developing and/or employing different regionalisation techniques. Previous experiences reveal that a skill for regional decadal predictions exists but that the interpretation of the results is quite complex due to their non-linear relationship to the global prediction skill. For example, Mieruch et al. (2014) found rather heterogeneous predictive skill for precipitation and temperature over Europe in the baseline0 generation. The skill differs over space, season, variable, and lead time after



initialisation. However, a general feature is an improved model spread for precipitation in the downscaled hindcasts when compared to their global counterparts. A potential for predicting regional peak winds and wind energy potentials over Central Europe several years ahead was identified in Haas et al. (2016) and Moemken et al. (2016). Particularly, they found highest skill scores for the first years after initialisation. However, these studies consider different variables, lead times, skill

metrics, and downscaling and data pre-processing methods, which makes it difficult to identify general conclusions for the decadal predictability over Europe in the MiKlip decadal prediction system.

In this study, the decadal predictive skill for temperature, precipitation, and wind speed over Europe is analysed for the baseline0 and baseline1 generation of the MiKlip system. With this aim, we used the same methodologies for all three variables to ensure comparability. Global MPI-ESM and downscaled hindcast ensembles are considered to address the

following three key questions:

- Is there a potential for skilful regional decadal predictions in Europe, and does this skill depend from the long-term trend?
- Does regional downscaling provide an added value for decadal predictions?
- How does the regional decadal predictive skill depend on the ensemble size?

The main topics of this paper are to demonstrate the potential of skilful regional decadal climate prediction for Europe for specific key climate variables to assess the added value compared to the respective global predictions and the impact of the ensemble size on the skill estimates. Thus, our focus lies on the methodological development and not on the physical mechanisms leading to the predictive skill.

The datasets used in this study are described in section 2, followed by the methodologies for data pre-processing and skill

analysis in section 3. The results for the three key questions are shown in section 4. Section 5 summarizes the results of a sensitivity analysis with respect to different pre-processing methodologies. A summary and discussion, as well as an outlook for future work are given in section 6.

## 2. Data

The analysed global hindcasts were simulated with the coupled model MPI-ESM in low-resolution (MPI-ESM-LR;

Giorgetta et al., 2013). Its atmospheric component is based on the ECHAM6 model (Stevens et al., 2013) with a horizontal resolution of T63 and 47 vertical levels, which is coupled to the MPIOM ocean model (Jungclaus et al., 2013) with a horizontal resolution of 1.5° and 40 vertical levels. Two hindcast generations are considered here, both computed with the MPI-ESM-LR but with different initialisation strategies. The first generation (baseline0; Müller et al., 2012) is initialised with oceanic conditions from an assimilation experiment, where the model state is nudged towards temperature and salinity

anomalies from NCEP/NOAA reanalysis (Kalnay et al., 1996). For the second generation (baseline1; Pohlmann et al., 2013b), temperature and salinity anomalies from the ocean reanalysis system 4 (ORAS4; Balmaseda et al., 2013) are used instead, together with a full-field 3-D atmospheric initialisation using fields from ERA40 (Uppala et al., 2005) and ERA-



Interim (Dee et al., 2011). For both generations, yearly initialised hindcasts are available, each of them comprising a 10-year period. For the downscaling experiment, global forcing for hindcasts of five starting dates are used (1 January 1961, 1971, 1981, 1991, and 2001; hereafter referred to as dec1960, dec1970, dec1980, dec1990, and dec2000) to cover the whole period from 1961-2010. For each starting date, an ensemble of 10 members was generated using 1-day lagged initialisation from the assimilation experiments (cf. Marotzke et al., 2016 for more details). This resulted in an ensemble of 50 global hindcasts per generation (baseline0 and baseline1; hereafter MPI_b0 and MPI_b1).

In this study, we analyzed global hindcasts dynamically downscaled to the EURO-CORDEX domain (Giorgi et al., 2006; cf Figure 1) at a horizontal grid resolution of 0.22° using the mesoscale non-hydrostatic regional climate model COSMO-CLM (CCLM; Rockel et al., 2008) on a rotated grid. The model version COSMO4.8-clm17 is employed. By using the MPI-ESM-LR ensemble as driving data, the global "initial condition" perturbation strategy is simply passed to the regional model. Analog to the global data, the experiment includes MPI_b0 and MPI_b1 downscaled hindcasts for dec1960, dec1970, dec1980, dec1990, and dec2000, with 10 members per decade (hereafter CCLM_b0 and CCLM_b1).

To evaluate the performance of both the global MPI-ESM and the regional CCLM hindcasts, a CCLM simulation run with reanalysis boundary conditions and observational datasets are used for verification. For temperature and precipitation we consider the observational dataset E-OBS (Haylock et al., 2008) based on the ECA&D (European Climate Assessment & Dataset; http://eca.knml.nl/) at a regular 0.25°x0.25° grid. As no gridded dataset is available for wind, a CCLM simulation forced with boundary conditions from ERA40 and ERA-Interim is employed as verification dataset for wind speed. For this reanalysis driven simulation CCLM is applied in the same model setup as for the regionalisation of the global hindcast ensemble (see above).

In this study, we want to quantify if the initialisation with observed climate states improves the performance of decadal predictions. To address this issue, uninitialised model simulations started from historical CMIP5 runs are usually considered as reference dataset (see also section 3.2). With this aim, a 10-member ensemble of uninitialised MPI-ESM-LR historical runs started from a pre-industrial control simulation are used, which are only forced by the aerosol and greenhouse gas concentrations for the period 1850-2005 (e.g. Müller et al., 2012).

## 3. Methods

### 3.1 Data processing

All datasets considered in this study are pre-processed in an analogous manner to enable a direct comparison. First, all data is interpolated to the same regular 0.25°x0.25° grid, which corresponds to the resolution of the E-OBS data. At each grid point, monthly anomaly time series are computed by subtracting the long-term means for the period 1961-2010 from the interpolated raw datasets.

In this study, we are primarily interested in anomalies on inter-annual to decadal timescales, which can be associated with the natural variability. Thus, to exclude responses to external radiative forcing, all monthly anomaly time series are de-





trended. We use a simple linear regression approach for all variables at each grid point to remove the long-term trend for the period 1961-2010, as the trend for the different variables over Europe cannot be uniformly defined but depends on the considered region (Christensen et al., 2007b). Finally, annual values are derived and multi-annual means for lead years 1-5 are built for further evaluation. In order to assess the impact of the de-trending on the predictive skill, the data processing

steps as described above are repeated without de-trending for a sensitivity study.

Following the suggestion of Goddard et al. (2013), the skill analysis is mainly performed for spatial means. Spatial averaging of the de-trended anomaly time series is performed for eight PRUDENCE regions over Europe (see Fig. 1; Christensen and Christensen, 2007a). Note that we only used grid points over land surfaces for the spatial means, as E-OBS data are not available over the oceans. Additionally, we calculated the predictive skill on the basis of all individual grid

points for specific exercises.

## 3.2 Skill metrics

The following three metrics are used to evaluate the performance of the global and regional hindcast ensembles and to address the three key questions: the continuous ranked probability skill score (CRPSS), the mean squared error skill score (MSESS), and the anomaly correlation coefficient (ACC). The skill metrics are applied to the pre-processed time series

described in section 3.1 and are computed for multi-annual means for lead time years 1-5 after initialisation. Recent studies analysing the MiKlip decadal prediction system demonstrated that the MiKlip ensemble performs best for the first years after initialisation for a wide range of variables, while the skill diminishes for longer forecast periods. For example, Müller et al. (2012) found highest skill scores for years 1-4 and 2-5 for annual mean surface temperature both for the North Atlantic region and global means. The same is true for annual wind speed and wind energy potentials over Central Europe, for which

skilful predictions are mainly restricted to the first years after initialisation (1-4 years), while negative skill scores are found for longer lead time periods (Moemken et al., 2016). Kruschke et al. (2014) provided evidence that the prediction skill for winter cyclones over the North Atlantic region is best for years 2-5 and reduced for longer time periods. Following the recommendation by Goddard et al. (2013), we focus in the following on the lead-time 1-5 years after initialisation, for which possible skill should originate mainly from the initialisation.

The CRPSS (e.g. Goddard et al., 2013) is often used to assess the reliability of probabilistic forecast models and defined as

$$CRPSS = 1 - \frac{CRPS_{hind}}{CRPS_{ref}}$$

with

$$CRPS = \int_{-\infty}^{\infty} [F(y) - F_o(y)]^2 \, dy$$

$CRPS_{hind}$ is the continuous ranked probability score (CRPS; Wilks, 2011), comparing cumulative distribution functions (CDFs) of the initialised hindcast experiments with CDFs of the verification dataset (observations). $CRPS_{ref}$ is the CRPS of a





reference dataset, which are in this study the uninitialized MPI-ESM-LR historical simulations. In case of a positive CRPSS the reliability in terms of the probabilistic quality of the forecast spread is higher in the initialised hindcasts than in the reference dataset, which is in this case the uninitialised historical ensemble. It can thus be used to test if the model ensemble spread adequately represents the forecast uncertainty.

The deterministic MSESS (Goddard, 2013) is defined as

$$MSESS = 1 - \frac{MSE_{hind}}{MSE_{ref}}$$

with

$$MSE = \frac{1}{N} \sum_{n}^{N} (\bar{X}_i - O_i)^2$$

where $MSE_{hind}$ is the mean squared error (MSE) between the ensemble mean of the initialised hindcasts ($X_i$) and the verification data, and $MSE_{ref}$ is the mean squared error of the uninitialised reference dataset versus the verification data ($O_i$). A positive MSESS means that the hindcasts are closer to the verification dataset than the uninitialised runs, indicating that

the initialisation leads to higher accuracy in predicting observed values. Note that independently from the ensemble size of the hindcast ensembles, the same historical 10-member ensemble is always used as reference dataset for the computation of CRPSS and MSESS.

The ACC (e.g. Wilks, 2011) is computed as the Pearson correlation between the ensemble mean of the hindcasts at a certain location i and the corresponding observations (Obs):

$$ACC_i = \frac{1}{N} \frac{\sum_t hind_t \, Obs_t}{\sigma_{hind} \, \sigma_{Obs}}$$

where $t = 1, ..., N$ is the time index. The ACC quantifies the accuracy of the predictions only in terms of the temporal course, while it is independent from the mean bias.

## 4. Results

### 4.1 Is there a potential for skilful regional decadal predictions in Europe?

In this section we address the key question of the general potential for skilful regional decadal predictions over Europe. With

this aim, we analyse both the potential added value of initialization compared to the (uninitialized) historical runs and the implications of removing the long-term trend for the predictive skill. Skilful in this context means an improvement of the skill metrics, e.g., when comparing de-trended decadal hindcasts to the uninitialised climate simulations. We therefore analyse the ability of the forecast system to better predict the climate variations up to five years ahead due to the initialization with observations using different skill metrics (see section 3.2). To determine the predictive skill over Europe,





the three skill scores were calculated for all individual land grid points of the Euro-CORDEX domain on the 0.25° grid (see section 3.1).

Fig. 2 shows MSESS plots for the de-trended time-series of temperature, precipitation and surface wind speed in CCLM_b0 and CCLM_b1. For temperature (Fig. 2a and 2b), positive skill scores are found in both ensembles over Scandinavia and for

the Mediterranean, while a stripe of negative values occurs over the British Isles and Central Europe. The largest deviations between CCLM_b0 and CCLM_b1 are found for Iberia, parts of southern France and Italy, where the MSESS is positive for CCLM_b1 but neutral to negative for CCLM_b0. To determine the effect of the trend on the predictive skill, we compare the data with (tr) and without trend (dtr) for the example of CCLM_b1 (first and second column of Table 1 and Table 2), keeping all other post-processing steps the same (see section 3.1). With trend included, the correlation improves (Table 1). It

shows high positive values between 0.68 - 0.96 in all regions, thus indicating that a predictive skill for temperature arises at least partially from a realistic prediction of the climate trend. The MSESS (Table 2) increases for all but the north-western regions (BI, FR, ME), but is less improved than the correlation.

Larger deviations between both ensembles are revealed for precipitation (Fig. 2c and 2d), where the MSESS fields are distinctly patchier when compared to temperature (Fig. 2a and 2b), reflecting the local character of rainfall. Both ensembles

show positive MSESS values for regions in Scandinavia, Eastern Europe, Iberia, and the British Isles (Fig. 2c and 2d). In CCLM_b1, predictive skill is also identified over Western Central Europe. Thus for CCLM_b1 positive skill is found for larger areas indicating an added value of the improved initialization procedure in baseline1 compared to baseline0. Like for temperature the skill scores for precipitation benefit from including the trends (column 3 and 4 of Table 1 and Table 2), although not as uniformly and strongly as for temperature. MSESS is improved only in the southern and eastern regions by

including the trend.

Regarding wind speed, the predictive skill in CCLM_b0 (Fig. 2e) shows high MSESS values over Scandinavia, Iberia, Southern Italy and along the coasts of the North and the Baltic Sea, while strongly negative values are found e.g. over most of France, southern Germany and the Alpine region. In CCLM_b1, the MSESS depicts low but positive values over most of Western and Central Europe, while strong negative values are now identified over parts of Eastern Europe (Fig. 2f). Overall

the predictive skill of CCLM_b0 is slightly higher and affects a larger area, indicating that the changes in the initialization method do not improve the results for wind speed. Including the trend leads to higher skill scores in most regions for CCLM_b1 (fifth and sixth column of Table 1 and Table 2). Here, the MSESS improves more distinctly than for the other variables.

We conclude that in terms of the MSESS accuracy there generally is a potential for skilful decadal predictions over Europe

in the regional MiKlip ensembles. However, the skill pattern is not uniformly found as it depends on the region and the variable, i.e. for individual regions the initialisation of the hindcasts and decadal predictions lead to an added value for accurate (retrospective) forecasts several years ahead, while for some regions the uninitialized historical runs deliver more reliable predictions. Further, the discrepancies between the two hindcast generations (CCLM_b0 and CCLM_b1), seem to indicate a slight shift in the pattern due to the different initialization methods for the global predictions.

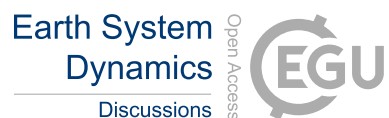

The skill of a prediction was also quantified using CRPSS and ACC. The spatial distribution of the CRPSS is very similar to that of the MSESS, while large deviations may arise for the ACC. This is exemplary shown in Fig. 3 for wind speed. For example, positive MSESS values are obtained for CCLM_b1 for Scandinavia and the coast of the North Sea (Fig. 2f), while the ACC is mainly negative for this domain (Fig. 3b). On the other hand, the spatial CRPSS patterns (Fig. 3c and 3d) agree

well to MSESS for both ensembles (Fig. 2e and 2f). Again, there are strong differences in the prediction skill for wind speed between CCLM_b0 and CCLM_b1, in particular for ACC (Fig. 3a and 3b). These differences might to some extent be associated with the different representation of the cyclone track density in the two ensembles. Kruschke et al. (2014) showed that the skill for winter cyclones is rather low in b1, while positive skill scores are detected in b0 over some parts of Europe and southeast of Iceland.

For the better understanding of the skill scores and their relation the different skill metrics are compared in scatter plots. Fig. 4a exemplary shows scatter diagrams of CRPSS vs the MSESS for temperature on individual grid point basis for CCLM_b1 for the mean over the lead-time 1–5 years. Generally, the accuracy and the reliability can vary highly with geographical position. However, and for the majority of the individual land grid points over Europe, positive MSESS are concurrent with positive CRPSS values (upper right quadrant). Both skill scores are linked to each other, as a quasi linear dependency

between CRPSS and MSESS is found. This is not only the case for CCLM_b1 (Fig. 4a) but also for MPI_b1 (Fig. 4b). In particular, we found that positive values for CRPSS often accompany with a high accuracy of the decadal predictions. This is generally true for all variables and both ensembles considered here (not shown).

On the other hand, no such linear relationship between ACC vs MSESS is found (see Fig. 4c for temperature). The ACC vs MSESS combination is clearly stronger scattered than for CRPSS vs MSESS, both in terms of the general spread and the

peak values of the number of grid points with a given skill score combination. Hence, a low mean bias of decadal predictions (resulting in positive MSESS values) does not necessarily imply a realistic temporal evolution. Still, positive MSESS values correspond to positive ACC values for most of the individual grid points, indicating a high potential for skilful regional decadal predictions over Europe. There are similar findings for precipitation (Fig. 4 e) with a broad distribution of the correlation values and a narrower range of the core area for the MSESS.

For ACC, keeping the original time series leads to enhanced predictability, while the impact of de-trending on the MSESS is less clear. In fact, the removal of a linear trend may in some cases be problematic. For example, if this trend is associated with a changing AMO phase, this may lead to an underestimation of the skill. During the investigation period, the AMO phase has indeed changed from cold to a warm (Sutton and Hodson, 2005). A proper attribution of the detected trends to greenhouse-gas induced climate change versus natural variability pattern is thus difficult.


## 4.2 Does regional downscaling provide an added value for decadal predictions?

Recent studies document that the application of regional climate models may improve climate simulations, in particular over complex terrain (Berg et al., 2013; Feldmann et al., 2013; Hackenbruch et al., 2016). This is mainly due to a more realistic



representation of the topography (e.g. mountain ranges or coast lines) in the RCMs compared to global-scale GCMs. In this section we analyse whether the downscaling with a regional climate model also leads to an added value for decadal predictions over Europe.

Figure 4 already indicates a shift of the overall distribution of skill scores from regionalised hindcasts towards higher values
compared to the global ones for the baseline1 ensemble. For temperature the core area of the skill values from the regional hindcasts (Fig 4a) is more confined to the upper right quadrant compared to the global ensemble (Fig 4b). This indicates an added value of downscaling for the accuracy as well as for the reliability. For the temperature correlation, the patterns are quite similar (Fig4 c, d), whereas for precipitation there is a clear shift towards an improved correlation and for a higher MSESS from the downscaling (Fig. 4 e,f). No or only a marginally low added value of regionalization on grid point scale is
observed for CCLM_b1 wind speed and for the majority of the variables and skill metrics in the baseline0 ensemble (not shown).

Ideally, an added value of downscaling should be accompanied by a positive absolute skill. Figure 5 depicts these two aspects for the three variables (2m temperature, precipitation, near-surface wind), the three verification metrics (MSESS, ACC, CRPSS), and the two ensemble generations (b0 and b1), as derived for the spatial means over the eight PRUDENCE
regions (cf. Fig. 1). Green dots indicate an added value of the CCLM results compared to MPI-ESM-LR and red dots no added value. Red background color indicates a negative skill score and green color a positive skill for the respective metric. This figure can be interpreted along several dimensions: (i) the skill for the different climate variables (background color), (ii) the improvement by downscaling (dot color), (iii) the improvement from b0 to b1, (iv) the skill for different regions, (v) and the different skill metrics.

For temperature, CCLM_b1 mostly shows an added value compared to MPI_b1 as well as compared to CCLM_b0. For most regions, this is particularly expressed in the MSESS and the anomaly correlation. For instance, with respect to the accuracy (MSESS and ACC) CCLM_b1 has higher skill in 6 of 8 PRUDENCE regions compared to MPI_b1. No added value of downscaling in both ensemble generations is found only for France (FR – region 3 from Fig. 1). Additionally, no benefit from downscaling could be detected with CCLM_b0 for the British Isles (BI – 1), Mid-Europe (ME - 4) and the
Mediterranean Area (MD - 7), where CCLM_b1 performs better. In general, in CCLM_b1 there are more regions with positive skill scores in southern Europe (IP - 2, AL - 6) and in Scandinavia (SC- 5). In the Mediterranean region both ensemble generations depict only positive skill scores.

For precipitation, an improvement from downscaling is detected particularly for CCLM_b1 over the majority of metrics and regions. In addition, CCLM_b1 is clearly superior to CCLM_b0 with respect to skill and added value. This indicates a
positive effect of the improved initialization procedures in b1 compared to b0 (Pohlmann, 2013b). However, this improvement does not affect all regions. CCLM_b0 performs better than its successor for the Iberian Peninsula, whereas skill and/or added value are higher in CCLM_b1 for the regions in the North-West (BI, FR, ME) and North (SC). With respect to the reliability CCLM_b1 outperforms CCLM_b0 for precipitation (CCLM_b0: 4 regions with positive CRPSS,



CCLM_b1: 7 regions with positive CRPSS), while for temperature both ensembles are equivalent (2 regions with positive CRPSS in CCLM_b0 and in CCLM_b1).

For the near surface wind Fig. 5 shows heterogeneous results. CCLM_b0 has an added value of downscaling in more regions than CCLM_b1. On the other hand CCLM_b1 provides an added value for the CRPSS in 4 regions, while for CCLM_b0 no added value of downscaling is found with respect to the reliability. CCLM_b0 has a positive skill in the Northern parts of the domain (BI, SC), whereas positive skill scores are found for CCLM_b1 over most other PRUDENCE regions at least for one skill metric. For Eastern Europe (EA - 8) none of the metrics are positive for both generations.

The detected shift in the skill patterns between CCLM_b0 and CCLM_b1 can be expected due to the different initialization procedures of the two generations. However, there also seem to be regions with more stable skill properties: The Mediterranean area shows positive skill for all variables and metrics (except wind in CCLM_b0).

An added value of regionalization over the majority of variables and metrics can be found for Southern Europe (MD, IP) and Scandinavia. As these areas have complex coastlines and orography, this result may be indicative of a better representation of small-scale processes in the CCLM. On the other hand, for the Alps (AL) only the ACC shows skill and added value from downscaling for temperature in both generations. The PRUDENCE region AL is the smallest of the regions, with the steepest orography. It might be that for the Alps an even higher resolution for the downscaling would be advantageous to improve the accuracy and reliability of the hindcasts.

We conclude that regional downscaling indeed may provide an added value for decadal predictions over Europe. However, while for some complex regions like MD, IP or SC this added value is to some extent systematic, for other areas in Europe the analysis reveals a mixed picture for the different variables and the skill metrics.

### 4.3 How does the regional decadal predictive skill depend on the ensemble size?

Past studies suggest that the ensemble size of a prediction system has an impact on the forecast skill of a model (Richardson, 2001; Ferro et al., 2008). Generally, there is consensus that the prediction skill for both seasonal and decadal predictions is enhanced when the number of ensemble members is increased. Kadow et al. (2014) analysed the global MiKlip baseline1 generation and concluded that the forecast accuracy for surface temperature for lead year 1 and 2-9 is improved for nearly the whole globe when the ensemble size is increased from 3 to 10 members. This is in line with the findings of Sienz et al. (2016), who examined the prediction skill for North Atlantic sea surface temperature in the same hindcast ensemble. Also for seasonal predictions of the North Atlantic Oscillation a forecast system profits from increasing size (e.g. Scaife et al., 2014). However, it is still open how a regional decadal forecast system does depend on the quantity of ensemble members. With this aim we analysed the impact of the ensemble size in the predictive skill for the eight PRUDENCE regions in Europe in both the regional and the global MiKlip ensembles. In the following, results are only exemplary shown for the Iberian Peninsula (IP), as the findings are similar for the other PRUDENCE regions. Figure 6 exhibits the dependency of CRPSS, MSESS, and ACC for lead years 1-5 (y-axis) on the ensemble size (x-axis) for all three variables spatially averaged over IP.



For each ensemble size $n$ ($n$ varying between 2 and 10), the solid coloured lines depict the averaged skill scores for all permutations of $n$-member ensemble combinations for each of the four individual hindcast ensembles (MPI_b0, MPI_b1, CCLM_b0, and CCLM_b1). Ranked probability skillscores may be negatively biased for small ensembles sizes (e.g. Ahrens and Walser, 2008), while such a bias is not reported for MSESS and ACC. To ensure a direct comparability of the results for

the three skill metrics we therefore decided not to use a de-biased version of the CRPSS in this study.

Enhanced predictive skill can be observed when the number of members is stepwise increased for both the global and the regional hindcast ensembles. MSESS and CRPSS show a rather logarithmic relationship with increasing n, depicting the highest skill scores for the 10 member ensembles for all three variables (Figure 6a-c and 6g-i). On the other hand, the lowest skill scores (often with negative values for CRPSS and MSESS) are always found for the 2-member ensembles. This

ensemble size dependency of MSESS and CRPSS is systematic and is found in both hindcast generations for all variables over all eight PRUDENCE regions (not shown), regardless whether the skill scores are negative or positive. In some cases, the ensemble size increase even leads to a shift from negative MSESS and CRPSS values to positive values (e.g. Fig. 6c, 6h, and 6i). In contrast, no systematic conclusion can be stated for the ACC, as the ensemble size dependency of the predictive skill depends on the variable and the considered MiKlip ensemble (Fig. 6d-f). But even here a larger ensemble size is

advantageous, as negative skill scores become more robust (cf. Fig. 6e, f). Nevertheless, there are also examples for the ACC where the ensemble size dependency is similar to that of MSESS and CRPSS, like e.g. for temperature (Fig. 6d). These results suggest that a decadal prediction system generally benefits from larger ensemble sizes, either in terms of more skilful and reliable decadal forecasts or at least of a reduction of the bias or the uncertainty, depending on the variable and the hindcast generation. Note that for most variables and skill scores the hindcast generation is more important for the skill than

the resolution. In additions, most diagrams indicate an added value of downscaling. For the reliability of wind speed both generations of CCLM surpass their MPI counterparts, indicating a systematic added value of downscaling.

For ensembles with less than 10 members, the skill scores of all possible $n$-member ensemble combinations are averaged. This is exemplary illustrated for the MSESS for precipitation in the CCLM_b0 ensemble (see box-whisker plots in Fig. 6b). While the spread between the individual $n$-member ensembles declines with an increasing number of members $n$, it is quite

large for small ensemble sizes: for instance, the MSESS varies between -1.5 and +0.8 for the 2-member ensembles (Fig. 6b). In fact, even for the 7-member ensemble quite different results can be found depending on the selection of the ensemble members, ranging from high positive MSESS values to zero. These results clearly demonstrate the necessity of using large ensembles to reduce these uncertainties.

We conclude that the predictive skill with respect to both accuracy and model spread is generally improved when the size of

the hindcast ensembles increases. This is valid for all variables, regions, and hindcast ensembles considered in this study. The skill scores converge towards a certain value in most cases for MSESS and CRPSS in all hindcasts (see Fig. 6a-c and 6g-i). The increments in added value by increasing the number of ensemble members decrease for more than 5 members. Nevertheless, it is recommended to use ten members or more for the skill assessment of decadal predictions on the regional scale.



## 5. Summary and discussion

In this study the decadal predictability in the regional MiKlip decadal prediction system is analysed for temperature, precipitation, and wind speed over Europe and compared to the forecast skill of the global ensemble. The goal is to assess the prospect of such a system for the application in forecasts on decadal timescales. Focus is given to years 1-5 after initialization. Three skill scores are used to quantify the accuracy and the reliability of the two different MiKlip hindcast generations. The main findings of our study can be summarized as follows:

- There is a potential for regional decadal predictability over Europe for temperature, precipitation, and wind speed in the MiKlip system, but the predictive skill depends on the variable, the region, and the hindcast generation.

- The MiKlip prediction system may distinctly benefit from regional downscaling. An added value in terms of accuracy and reliability is particularly revealed for temperature over the British Isles (BI), Scandinavia (SC), the Mediterranean (MD), and for precipitation over the British Isles (BI), Scandinavia (SC), Mid-Europe (ME), and France (FR) for the b1 generation. Most of these regions are characterized by complex coastlines and orography, which indicates that the better representation of topographic structures in the regionalised hindcasts may improve the predictive skill.

- The improvement of the initialization procedure from baseline0 to baseline1 as described in Pohlmann et al. (2013b) increases the overall predictive skill in the downscaled MiKlip hindcasts over Europe, at least for precipitation and temperature. But improvement of the skill varies between variable and region. The skill for temperature increases around the Mediterranean Sea and parts of Scandinavia from b0 to b1. For precipitation the skill of b1 compared to b0 is higher in all regions but the Iberian Peninsula. Only for wind speed there is mostly no benefit from the improved initialization in most regions.

- A systematic enhancement of MSESS and CRPSS skill scores is found with increasing ensemble size, and a number of 10 members is found to be suitable for decadal predictions. This is valid for all variables and European regions in the global and regional MiKlip ensembles.

- The predictive skill may increase when keeping the original time series including the long-term trend. A linear de-trending may remove parts of the signal since the climate trend and the AMO teleconnection pattern are in phase both contributing to ascending trends over the hindcast period 1960 – 2010.

Müller et al. (2012) and Pohlmann et al. (2013b) had found systematic prediction skills for surface temperature over large parts of the North-Atlantic and Europe in both global generations (baseline0, baseline1). From the results of our study, it is apparent that key European regions for decadal predictability (beyond the climate trend) with the regional prediction system seem to be the Mediterranean Area and the Iberian Peninsula. This is in line with findings from Guemas et al. (2015). This finding may be related with skilful predictions of the AMO (Garcia-Serrano et al., 2012; Guemas et al., 2015). Due to the rather non-linear relationship of these large-scale North Atlantic features to regional atmospheric conditions over Europe, the mechanisms steering the decadal variability and predictability of climate variables in European regions are thus more



complex. The decadal variability of regional precipitation, temperature, and wind speed over most parts of Europe is largely affected by the North Atlantic oscillation, but its skilful decadal predictability over the continent is still under debate. With this respect, a better understanding of the mechanisms relevant for the regional climate over Europe on the decadal time scale is required, as was for example obtained for the tropical Atlantic (Dunstone et al., 2011). This is an objective of the

ongoing 2nd phase of the MiKlip project.

The skill scores may strongly vary between neighbouring grid points. Comparable results were found by e.g. Guemas et al. (2015), who detected a rather diffuse pattern for the accuracy of decadal predictions over Europe for seasonal temperature and precipitation. This might at least partly be due to spatial and temporal inhomogeneity of the gridded observational references. A more realistic assessment of the prediction skill can be made by considering spatial means (Goddard et al.,

2013) which was mostly considered in this study. In line with e.g. Kadow et al. (2016), we could show that an enlargement of the ensemble size up to 10 members results in an improvement of the prediction skill over Europe. However, prediction skill could further benefit from even larger ensemble sizes, especially in areas with low signal-to-noise ratio (cf. Sienz et al., 2016).

Bias and drift adjustment (e.g., Boer et al., 2016) provide prospect in skill improvement not only for GCMs but also for

RCMs. This is particularly the case for ensemble simulations run with full-field initialization (prototype, not analysed here; cf. Marotzke et al., 2016). While bias and drift adjustment methods have improved the forecast skill of near-term climate prediction (e.g., Kruschke et al., 2016), such corrections are less important for the baseline0 and baseline1 ensembles analysed here as they were generated with anomaly initialisation (Marotzke et al., 2016). Nevertheless, bias correction and calibration are an important topic in the second phase of MiKlip.

Due to the high computational costs of dynamical downscaling, only five starting dates (one per decade) are available for the regional MiKlip ensemble generation b0 (see section 2). This is a shortcoming regarding the statistical significance of the results and some of the statements presented in this study. The statistical significance will be easier to quantify when the regional simulations for the newest Miklip ensemble generation are available with annual starting dates over more than 50 years. On the other hand, regional decadal forecasts may have advantages beyond the examples discussed in this paper. For

example, RCMs enables the integration of improved components of the hydrological cycle or climate-system components with memory on multi-year time-scales like soil moisture (Khodaya et al., 2014; Sein et al., 2015). Kothe et al. (2016) has shown that extracting the initial state of the deep soil in the RCMs from regional data assimilation schemes may improve decadal predictions. Further, Akhtar et al. (2017) demonstrated that the regional feedback between large water bodies and the atmosphere play a major in the regional climate system. This feedback can only be captured in regionalized climate

predictions by a dynamic RCM-ocean coupling. Most of the approaches mentioned above are ongoing within the 2$^{nd}$ phase of MiKlip and are expected to enhance the decadal predictability over Europe. We thus conclude that a decadal prediction system would clearly benefit from a regional forecast ensemble.

The regional decadal prediction system generated by the MiKlip consortium comprises altogether 1000 years (two hindcast generations, each of them comprising ten hindcast members for five starting years) of simulations with 0.22° for the entire




EURO-CORDEX region, which is a to our best knowledge unprecedented. Hence, this ensemble enabled us to gain important insights into different aspects and the prospects of regional downscaling for decadal predictions, and serve as a good basis for future studies. In the ongoing 2$^{nd}$ phase of MiKlip it is planned to downscale a complete ensemble hindcast generation with ten members for more than 50 starting years, giving altogether more than 5000 years.

## Author Contributions

MR, HF, SM and MU developed the concept of the paper; MR, HF and JGP wrote the first manuscript draft. MR, HF, SM, MU, NL and JM contributed with data analysis and analysis tools. HF, SM, MR, BA, BF contributed with RCM simulations. MK and WM contributed with the global MPI-ESM-LR simulations and prepared boundary conditions for RCM simulations. CK leads the MiKlip-C consortium, with CO-Is BA, BF, JGP, GS. All authors contributed with ideas,

interpretation of the results and manuscript revisions.

## Acknowledgments

MiKlip is funded by the German Federal Ministry for Education and Research (BMBF, contracts: 01LP1518 A-D and 01LP1519) All simulations were carried out at the German Climate Computing Centre (DKRZ), which also provided all major data services. We acknowledge the E-OBS data set from the EU-FP6 project ENSEMBLES (http://ensembles-eu.metoffice.com) and the data providers in the ECA&D project (http://www.ecad.eu). We thank the European Centre for Medium-Range Weather Forecasts (ECMWF) for their ERA-40 and ERA-Interim Reanalysis data (http://apps.ecmwf.int/datasets/). JGP thanks the AXA Research Fund for support. We thank past and present members of the MiKlip –C (Regionalization) group for discussions and comments.

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



**Figures**

**Figure 1: CCLM modelling domain (= EURO-CORDEX domain): Modell orography and PRUDENCE regions. 1: British Isles BI; 2: Iberian Peninsula IP; 3: France FR; 4: Mid-Europe ME; 5: Scandinavia SC; 6: Alps AL; 7: Mediterranean MD; 8: Eastern Europe EA.**



**Figure 2: Spatial distribution of the MSESS for the multi-annual mean of lead years 1-5 for (a) temperature in CCLM_b0, (b) temperature in CCLM_b1, (c) precipitation in CCLM_b0, (d) precipitation in CCLM_b1, (e) wind speed in CCLM_b0, and (f) wind speed in CCLM_b1. All datasets have been de-trended, and as reference dataset we have used the uninitialized historical ensemble.**





**Figure 3: Spatial distribution of the skill scores for the multi-annual mean of lead years 1-5 for wind speed. (a) ACC for CCLM_b0, (b) ACC for CCLM_b1, (c) CRPSS for CCLM_b0, and (d) CRPSS for CCLM_b1. All datasets have been de-trended, and for CRPSS we have used the uninitialized historical ensemble as reference dataset.**





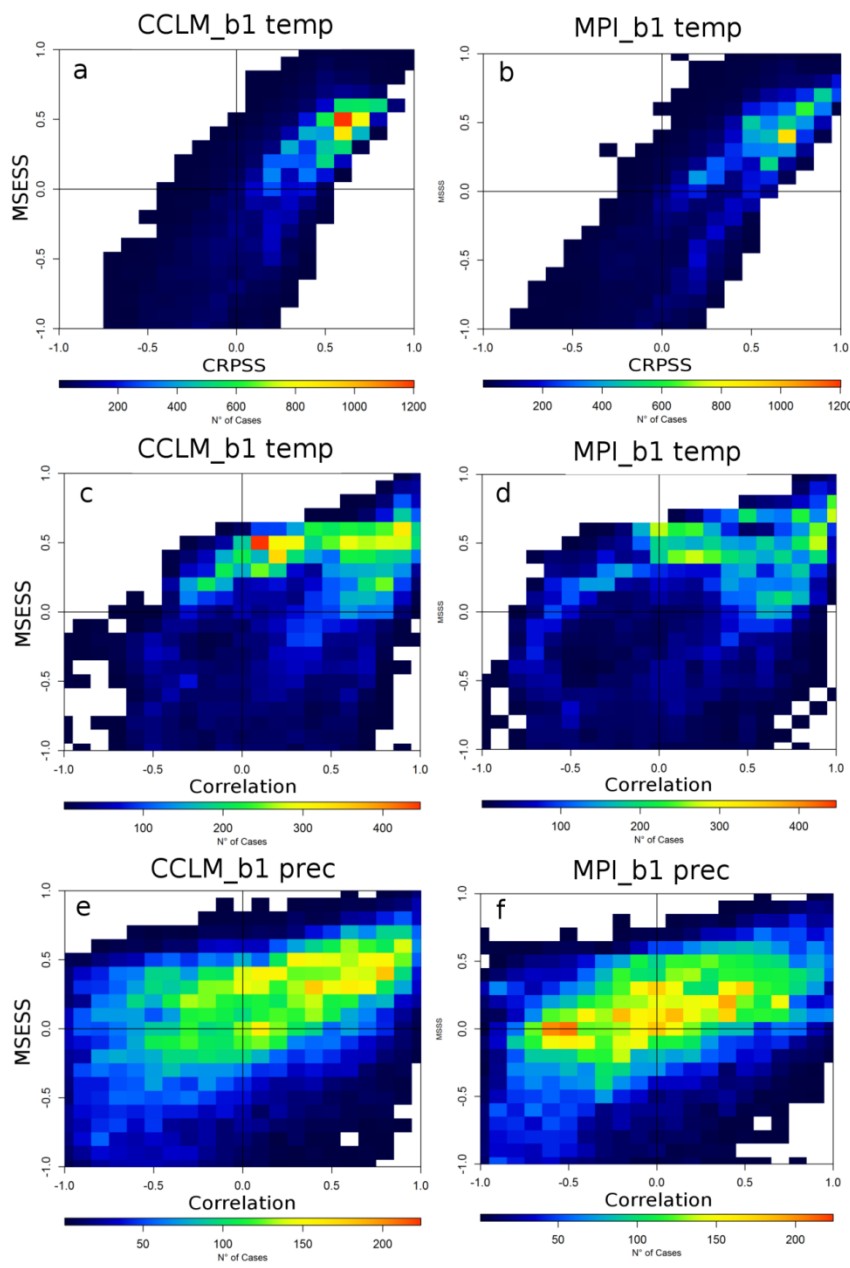

**Figure 4: Scatter diagrams for CRPSS (x-axis) vs MSESS (y-axis) for temperature at all individual EURO-CORDEX grid points for the multi-annual mean of lead years 1-5 in (a) CCLM_b1 and (b) MPI_b1. (c), (d) as (a), (b) but for ACC vs MSESS for temperature. (e), (f) as (a), (b) but for ACC vs MSESS for precipitation. Colours denote the number of grid points over Europe with a given skill score combination. All datasets have been de-trended, and for MSESS and CRPSS we have used the uninitialized historical ensemble as reference dataset. Note the different scaling of the colour bars.**



**Figure 5: Predictive skill (MSESS, ACC and CRPSS) and added value of the regional MiKlip ensembles (CCLM_b0 and CCLM_b1) over the eight PRUDENCE regions (cf. Fig. 1) for temperature (left columns), precipitation (middle), and 10m-wind (right) for the multi-annual mean of lead years 1-5. Red filled boxes indicate negative skill scores, green filled boxes positive skill scores. Green dots denote an added value compared to the global forcing by MPI-ESM-LR, red dots indicate no added value by regionalization. All datasets have been de-trended, and for MSESS and CRPSS we have used the uninitialized historical ensemble as reference dataset.**





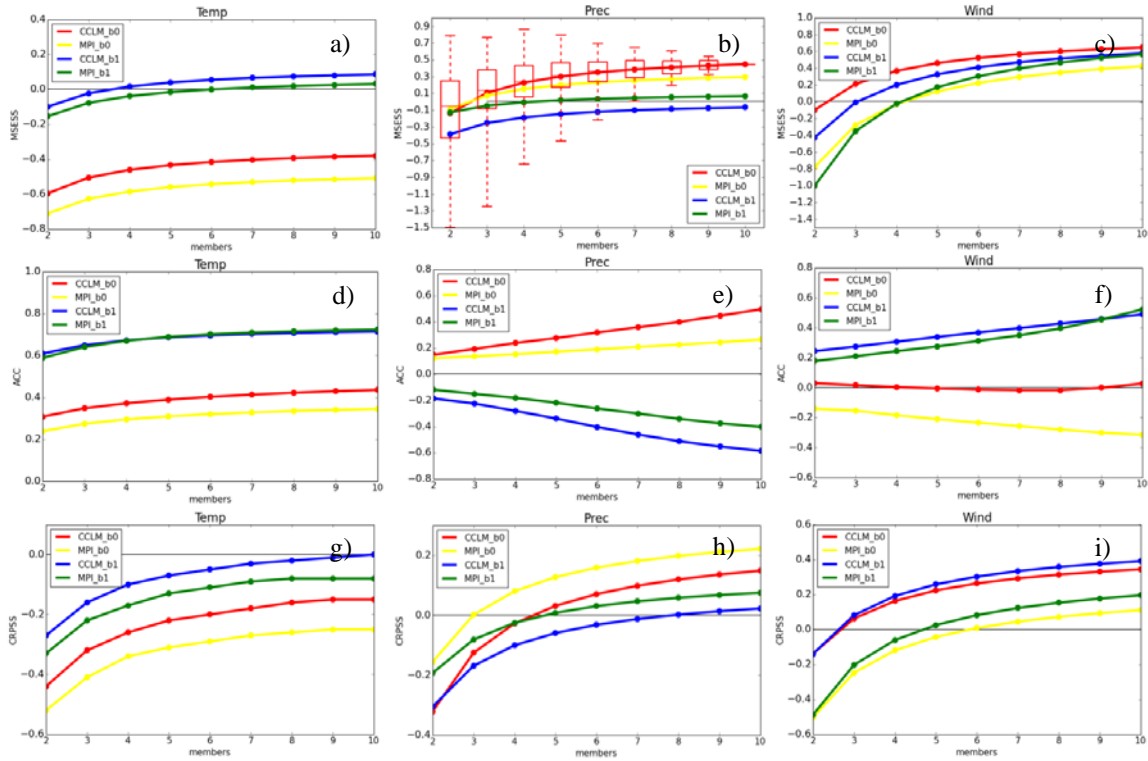

**Figure 6: Skill scores for the multi-annual mean of lead years 1-5 of the CCLM_b0 (red), MPI_b0 (yellow), CCLM_b1 (blue), and MPI_b1 (green) ensembles depending on the ensemble size (x-axis, ranging from 2 to 10 members) over IP (cf. Fig. 1). MSESS for (a) temperature, (b) precipitation, and (c) wind speed; ACC for (d) temperature, (e) precipitation, and (f) wind speed; CRPSS for (g) temperature, (h) precipitation, and (i) wind speed. In (b) box-whisker plots for the skill scores of all n-member combinations are shown. All datasets have been de-trended, and for MSESS and CRPSS we have used the uninitialized historical ensemble as reference dataset. Note the different scaling of the y-axis. For details please refer to main text.**





**Tables**

| | Temperature | | Precipitation | | Wind | |
|---|---|---|---|---|---|---|
| | **dtr** | **tr** | **dtr** | **tr** | **dtr** | **tr** |
| **1 BI** | -0.18 | **0.68** | 0.27 | **0.49** | -0.54 | -0.57 |
| **2 IP** | 0.71 | **0.87** | -0.59 | **0.19** | 0.49 | **0.63** |
| **3 FR** | 0.69 | **0.92** | 0.77 | 0.50 | 0.15 | **0.32** |
| **4 ME** | -0.12 | **0.80** | 0.64 | **0.79** | -0.37 | **0.04** |
| **5 SC** | 0.44 | **0.69** | -0.07 | **0.63** | -0.63 | -0.54 |
| **6 AL** | 0.83 | **0.97** | 0.37 | 0.18 | 0.10 | **0.44** |
| **7 MD** | 0.95 | **0.96** | 0.55 | **0.91** | 0.48 | 0.42 |
| **8 EA** | -0.09 | **0.73** | -0.10 | -0.04 | -0.94 | -0.39 |

**Table 1: ACC for temperature, precipitation, and wind speed in the CCLM_b1 ensemble over all eight PRUDENCE regions (cf. Fig. 1) for lead-years 1-5. De-trended 5-year averages (dtr), and 5-year averages with retained trend (tr). Lower skill scores in tr compared to dtr are marked in blue. Higher skill scores in tr compared to dtr are marked in red, and if skill scores of tr are additionally positive they are marked in bold red and underline. The uninitialized historical ensemble has been used as reference dataset. For details see main text.**



| | Temperature | | Precipitation | | Wind | |
|---|---|---|---|---|---|---|
| | dtr | tr | dtr | tr | dtr | tr |
| **1 BI** | -4.58 | -6.94 | 0.15 | 0.15 | -0.54 | -0.42 |
| **2 IP** | 0.09 | 0.11 | -0.07 | -0.49 | 0.58 | **0.84** |
| **3 FR** | -0.58 | -2.07 | 0.44 | 0.34 | 0.21 | **0.56** |
| **4 ME** | -1.46 | -2.40 | 0.44 | **0.63** | 0.18 | 0.14 |
| **5 SC** | 0.15 | **0.32** | 0.26 | 0.21 | 0.10 | 0.09 |
| **6 AL** | 0.44 | **0.47** | -0.36 | -0.18 | -0.23 | **0.19** |
| **7 MD** | 0.68 | **0.76** | 0.68 | **0.73** | 0.20 | **0.43** |
| **8 EA** | -1.78 | -0.42 | 0.31 | **0.36** | -0.72 | **0.22** |

**Table 2: As Table 1, but for MSESS.**

