# Peer review of "Development and prospects of the regional MiKlip decadal prediction system over Europe: Predictive skill, added value of regionalization and ensemble size dependency"

_Earth System Dynamics, 2017_

## Referee Comment (RC1) · Anonymous Referee #1 · 10 Oct 2017

General comments This is a very good study that focuses on the potential merits of regional downscaling decadal climate predictions over Europe. Specifically, the MiKlip prediction system studied uses the low resolution MPI global decadal hindcast ensemble at T63 resolution and dynamically downscales these hindcasts over Europe using the COSMO-CLM model at $0.22°$ horizontal resolution. Two 10 member ensemble regional hindcasts of 5 start dates are examined and verified against observational analyses of surface temperature, precipitation and low level wind using three different skill metrics, MSESS, CRPSS and ACC. The authors examine these metrics to answer

the following questions: is there potential for skillful regional predictions in Europe? Does regional downscaling provide added value? and How does the skill of these predictions depend on ensemble size? The first two questions are answered affirmatively and for the last question ensemble size stabilizes the skill metrics MSESS and CRPSS at ten members but ACC skill depends on ensemble size beyond ten members. The manuscript meets all the criteria for publication and needs only minor changes.

Specific comments The manuscript could be improved in two ways that would increase the significance of the work. First, although there are significantly large regions in Europe where the skill of the initialized hindcasts is positive, there is also a large region in central Europe where the skill is negative. This is particularly true of the MSESS of temperature. Since the reference is forecast is an uninitialized ensemble of 20th century simulations this raises the question as to the reason for this negative skill. The answer or some speculation to how it arises should be included in the article. In a similar vein, the authors do not include in their discussion any metrics that use The observed climatological distribution as the reference forecast, so that skill is measured solely using comparison with observations.

Technical corrections

Pg 2 Yaeger et al should be Yeager et al Pg 8 stronger scattered should be more strongly scattered

---

## Referee Comment (RC2) · Anonymous Referee #2 · 19 Dec 2017

The paper gives a preliminary assessment of regional decadal prediction skill over Europe based on a high-resolution regional model forced with boundary conditions obtained from the low-resolution, global MiKlip prediction system. I deem the analysis preliminary because the "development and prospects" of the downscaling system are being assessed at a rather early stage when only 5 hindcast start dates have been completed using the regional model. This is a serious shortcoming that calls into question the reliability of skill scores (computed from 5 data pairs) that are used throughout to make statements about the benefits of downscaling for various fields in various Eu-

ropean regions. Two-tier decadal prediction involving regional downscaling is certainly a topic of high interest, but this manuscript has the feel of an internal technical note that documents some preliminary and very mixed results that are still clouded in uncertainty given the limited temporal sampling. Unless it can be shown (perhaps using the MPI baseline systems) that 5 start dates are sufficient to get an accurate estimate for the skill scores and fields of interest, then what is the point of all this? I suspect that 5 start dates is not sufficient, and that the skill scores reported here are very "noisy" as a result. This may contribute to the mixed results and lack of strong take-away messages from this paper. It may be better to wait until more downscaled start dates have been completed before resubmission of this analysis.

Another main concern is the use of detrending, which probably exacerbates the sampling issues (how well-defined is a trend computed from 5 data points?). There is no real need to detrend since you have an uninitialized ensemble that allows you to determine the skill improvement relative to the externally-forced signal (yes, pure ACC will be higher, but you can show delta(ACC), i.e. the change in ACC relative to the uninitialized ensemble).

The quality of the writing is decent, but not high, and there are numerous instances of poor English construction (some noted below). A thorough proofreading is in order if this is to be resubmitted.

Specific Comments and Questions:

P2,L8: Here and throughout: "Yaeger" should be "Yeager".

P2,L11: It's not clear what the point is of the "while few" construction. Are you contrasting the large number of studies focusing on global metrics with the relatively few studies focusing on storm tracks, etc? Please rewrite.

P2,L13: What is this an example of? Why cite Sutton and Hodson (2005) in a paragraph focused on initialized decadal prediction?

P3,L7-18: The motivation for the present work needs to be clarified, particularly since it is not at all clear how the present study differs from the closely related recent MiKlip studies that have just been cited (Kadow et al. 2016; Mieruch et al. 2014; Haas et al. 2016; Moemken et al. 2016).

P3,L11: This question is poorly phrased. Do you mean "depend on" the trend or "derive from" the trend?

P3,L15: This is a repetitive rephrasing of the questions just covered.

P3,L29: I don't understand how ocean temperature and salinity can be nudged towards NCEP/NOAA reanalysis, since the latter is an atmospheric reanalysis.

P4,L11: Not clear what is meant by "Analog to the global data"?

P4,L14: Replace "are" with "is".

P4,L16-19: You already introduced the ERA-driven CCLM simulation in the first line of this paragraph, so consolidate your sentences into one brief description.

P4,L21: I don't understand what you mean by "uninitialized model simulations started from historical CMIP5 runs". Do you mean downscaled simulations that can be considered "uninitialized" counterparts to CCLM_b0 and CCLM_b1? Do you mean "pre-industrial CMIP5 runs"?

P4,L32: Replace "the natural variability" with "natural variability". Why use linear detrending to isolate natural variability when you have just introduced an uninitialised ensemble that can be used to quantify the skill associated with external forcing?

P5,L14: I think you mean "post-processed time series".

P5,L24: What is the basis for claiming that "skill should originate mainly from the initialization" as opposed to the external forcing? This has not been shown and shouldn't be assumed.

[Figure]

P5,L25-: What are F(y) and Fo(y)? Please explain the CRPS equation. What exactly is CDF and how is it computed?

P6,L23-P7,L2: This is repetitive.

Fig 2: Suggest using a nonlinear scale for MSESS, such as -9 to 0.9 as in Shaffrey et al. (2016, doi:10.1007/s00382-016-3075-x), because this metric is not symmetric about 0 in terms of relative improvements in MSE. Please clarify that these are for annual mean (ie, not seasonal mean) anomalies.

P7,L7: I presume the detrending has been performed similarly for observations and for the uninitialized historical runs? This isn't explicitly mentioned.

Table 1: What is the meaning of "The uninitialized historical ensemble has been used as reference dataset", given that this is a table of ACC scores? Am I correct that this table displays correlations computed from 5 data points (corresponding to the 5 start years)? Clearly the externally-forced trend is important and so this table should include ACC scores for the uninitialized historical runs for comparison.

P7,L11: What is the meaning of "increases"—relative to uninitialized or relative to detrended?

P7,L34: I would say Figure 2 shows more than a "slight shift".

P8,L2: Here and elsewhere delete "exemplary" as it is not being used properly.

P8,L4: It's curious that Fig 2e agrees so well with Fig 3a, but Fig 2f is so different from Fig 3b. Can you offer any explanation? In my mind, it calls into question the significance of skill scores computed from 5 data points.

P8,L25-29: This discussion begs the question of why you are doing any detrending at all (see comment above)? The purpose of the uninitialized ensemble is precisely to allow you to discriminate between greenhouse-gas induced variability (including trends) and natural variability (including AMO-related trends). Detrending is confusing matter-

s—just compared initialized to uninitialized skill.

P9,L8: Change "whereas" to "and".

P9,L9-11: This incomprehensible sentence needs a rewrite.

P11,L5: I don't understand this sentence.

P11,L24-28: This is because you are doing bootstrapping without replacement; if you allow replacement, then the spread does not necessarily diminish with ensemble size.

---

## Author Comment (AC1) · 16 Jan 2018

Reviewer 1

General comments
This is a very good study that focuses on the potential merits of regional downscaling decadal climate predictions over Europe. Specifically, the MiKlip prediction system studied uses the low resolution MPI global decadal hindcast ensemble at T63 resolution and dynamically downscales these hindcasts over Europe using the COSMO-CLM model at 0.22_ horizontal resolution. Two 10 member ensemble regional hindcasts of 5 start dates are examined and verified against observational analyses of surface temperature, precipitation and low level wind using three different skill metrics, MSESS, CRPSS and ACC. The authors examine these metrics to answer the following questions: is there potential for skillful regional predictions in Europe? Does regional downscaling provide added value? and How does the skill of these predictions depend on ensemble size? The first two questions are answered affirmatively and for the last question ensemble size stabilizes the skill metrics MSESS and CRPSS at ten members but ACC skill depends on ensemble size beyond ten members. The manuscript meets all the criteria for publication and needs only minor changes.

Specific comments:
The manuscript could be improved in two ways that would increase the significance of the work. First, although there are significantly large regions in Europe where the skill of the initialized hindcasts is positive, there is also a large region in central Europe where the skill is negative. This is particularly true of the MSESS of temperature. Since the reference is forecast is an uninitialized ensemble of 20[th] century simulations this raises the question as to the reason for this negative skill. The answer or some speculation to how it arises should be included in the article. In a similar vein, the authors do not include in their discussion any metrics that use the observed climatological distribution as the reference forecast, so that skill is measured solely using comparison with observations.

Answer: We thank the Reviewer for these helpful comments, which will certainly help to improve our manuscript. There are several potential reasons for the negative skill for temperature, including the detrending of the time series as queried by Reviewer 2. Following the suggestions of Reviewer 2 we will redo most of the analysis without detrending, which likely will result in new skillscore plots. The comparison of the new plots with those shown in the manuscript will probably help to better understand the reason for the negative skills, and we will discuss this in detail in the revised version when the new plots are available.
Further, we followed the suggestion and performed an additional analysis including the climatology as reference. See for instance the upper row plots in Fig. A1-A3 attached to the answers to Reviewer 2 for ACC, which show the correlation w.r.t. the observations. The other skill scores (MSESS and CRPSS) have also been calculated for both references, namely the historical ensemble and the climatological distribution.  We would include this information in the revised paper.
In Fig A1 you can see, that the skill in Central Europe is high when using the climatology as reference forecast (Fig. A1 upper left). But, in this region the historical ensemble shows an even higher skill (Fig.A1 lower left). Part of the reduced skill there can be attributed to the low sample size. Fig. A1 lower right indicates less negative to slightly positive skill scores compared to the historical ensemble when a larger sample size with annual starting values are used. We intend to include this additional analysis  in the paper (see also answers to Reviewer 2).

Technical corrections
Pg 2 Yaeger et al should be Yeager et al
A: This will be corrected

Pg 8 stronger scattered should be more strongly scattered
A. We will change it accordingly

---

## Author Comment (AC2) · 16 Jan 2018

Reviewer 2

Comment of the authors:

We thank the Reviewer for his/her thoughtful review and the specific suggestions. The Reviewer queried two major points: (i) the low number of starting dates used in our study (which has been discussed in detail in the original manuscript) and (ii) the detrending of the time series. As we generally agree with the Reviewer in both points and take the concerns seriously, we decided to include new Figures in our responses with respect to point (i) (see also answer to main concerns below), and to redo most of the calculations without detrending the time series as suggested by the Reviewer and replace the respective Figures in the revised manuscript. As the new calculations will impact the majority of our results quantitatively, a reformulation of most parts of the manuscript will be necessary. Therefore, in most of our responses to the specific comments (see below) we only stated which changes will be made, without giving the manuscript changes explicitly at this point in time. The point-to-point responses to all major and specific comments are marked in red.

**Anonymous Referee #2**

*The paper gives a preliminary assessment of regional decadal prediction skill over Europe based on a high-resolution regional model forced with boundary conditions obtained from the low-resolution, global MiKlip prediction system. I deem the analysis preliminary because the "development and prospects" of the downscaling system are being assessed at a rather early stage when only 5 hindcast start dates have been completed using the regional model. This is a serious shortcoming that calls into question the reliability of skill scores (computed from 5 data pairs) that are used throughout to make statements about the benefits of downscaling for various fields in various European regions.*

*Two-tier decadal prediction involving regional downscaling is certainly a topic of high interest, but this manuscript has the feel of an internal technical note that documents some preliminary and very mixed results that are still clouded in uncertainty given the limited temporal sampling. Unless it can be shown (perhaps using the MPI baseline systems) that 5 start dates are sufficient to get an accurate estimate for the skill scores and fields of interest, then what is the point of all this? I suspect that 5 start dates is not sufficient, and that the skill scores reported here are very "noisy" as a result. This may contribute to the mixed results and lack of strong take-away messages from this paper. It may be better to wait until more downscaled start dates have been completed before resubmission of this analysis.*

Answer:

We agree that five starting years lead to "noisy" results. But we argue, that this noisiness affects chapter 4.1 the most, dealing with the skill distribution over Europe. The chapters 4.2 regarding the added value and chapter 4.3 regarding the ensemble size provide robust results even when using two times 50 simulations. Therefore, the major parts of the results are not affected strongly by the sample size issue.

To remedy the sample size and starting year issue we performed, as the reviewer suggested a comparative analysis of the skill estimates for the three variables addressed in the paper derived from a) starting years every 10 years (1960, 1970,..,2000) as in the original

manuscript and b) annual starting dates (1960-2005) for the global 10 member ensemble with MPI-ESM-LR baseline1. Baseline1is the only ensemble used in the paper which provides 10 members throughout the whole hindcast period. As reference we applied 10 member of the un-initialized "historical" ensemble with MPI-ESM-LR. The results (see figures for the correlation at the end of the document) show a general qualitative agreement, though of course not a quantitative one. As expected, larger sample size provides smoother skill estimates, less noisy than with the smaller sample size. But in general the findings regarding the regions with hindcast skill mentioned in the original manuscript are still correct for the extended ensemble. We offer to include this additional analysis in the paper to point out, how and where the smaller sample size affects the findings and that way putting them into perspective.

*Another main concern is the use of detrending, which probably exacerbates the sampling issues (how well-defined is a trend computed from 5 data points?). There is no real need to detrend since you have an uninitialized ensemble that allows you to determine the skill improvement relative to the externally-forced signal (yes, pure ACC will be higher, but you can show delta(ACC), i.e. the change in ACC relative to the uninitialized ensemble).*
*The quality of the writing is decent, but not high, and there are numerous instances of poor English construction (some noted below). A thorough proofreading is in order if this is to be resubmitted.*

A: We thank the Reviewer for this helpful comment. We agree that detrending is not necessary when using an uninitialized ensemble as reference. We therefore decided to redo most of the analysis of our study without detrending of the time series and include the new results in the revised manuscript. The figures attached at the end of this document (see also the response to the first main concern) show the ACC for the baseline1 generation and additionally the difference to the ACC of the uninitialized historicals as suggested by the Reviewer. The respective Figures without detrendeing for the other skill metrics will be included in the revised manuscript. Further, we will proofread the revised version of the manuscript as suggested by the Reviewer.

Specific Comments and Questions:
P2,L8: Here and throughout: "Yaeger" should be "Yeager".
A: Citation will be changed throughout in the revised version.

P2,L11: It's not clear what the point is of the "while few" construction. Are you contrasting the large number of studies focusing on global metrics with the relatively few studies focusing on storm tracks, etc? Please rewrite.
A: We agree with the Reviewer that this sentence is misleading. We will rephrase it in the revised version.

P2,L13: What is this an example of? Why cite Sutton and Hodson (2005) in a paragraph focused on initialized decadal prediction?

A: This sentence will be removed in the revised manuscript.

P3,L7-18: The motivation for the present work needs to be clarified, particularly since it is not at all clear how the present study differs from the closely related recent MiKlip studies that have just been cited (Kadow et al. 2016; Mieruch et al. 2014; Haas et al. 2016; Moemken et al. 2016).
A: As stated in line 4-6 on page 3, the closely related MiKlip studies are difficult to compare, as they use different skill metrics, pre-processing methods and downscaling approaches. The unique feature of our study is that we use the same methods/metrics not only for the

regional but also for the global prediction system, which enables us to give a more general assessment of the prospects of the MiKlip system with respect to near surface variables which affect human life most (temperature, precipitation, and wind speed). However, we agree that this sentence is not sufficient to emphasize our motivation. We will rephrase it in the revised version.

P3,L11: This question is poorly phrased. Do you mean "depend on" the trend or "derive from" the trend?
A: We agree that this question is misleading. We will remove the second part of the question.

P3,L15: This is a repetitive rephrasing of the questions just covered.
A: Paragraph in line 15-18 will be removed in the revised manuscript.

P3,L29: I don't understand how ocean temperature and salinity can be nudged towards NCEP/NOAA reanalysis, since the latter is an atmospheric reanalysis.
A: The reviewer is correct. In baseline0 the ocean salinity and temperature anomalies were derived from a simulation with the ocean model MPI-OM forced with the NCEP re-analysis. We will change the sentence accordingly.

P4,L11: Not clear what is meant by "Analog to the global data"?
A: To avoid misinterpretation, we will change the text:
"The experiment includes downscaled hindcasts for dec1969, dec19790, dec1980, dec1990, and dec2000, with ten members per decade (hereafter CCLM_b0 and CCLM_b1). The regional ensembles therefore consist of the same time series like the global ensembles MPI_b0 and MPI_b1."

P4,L14: Replace "are" with "is".
A: Will be replaced in the revised manuscript.

P4,L16-19: You already introduced the ERA-driven CCLM simulation in the first line of this paragraph, so consolidate your sentences into one brief description.
A: We will change the first sentence of this paragraph:
"To evaluate the performance of both the global MPI-ESM and the regional CCLM hindcasts, reanalysis and observational datasets are used for verification."

P4,L21: I don't understand what you mean by "uninitialized model simulations started from historical CMIP5 runs". Do you mean downscaled simulations that can be considered "uninitialized" counterparts to CCLM_b0 and CCLM_b1? Do you mean "preindustrial CMIP5 runs"?
A: We agree that this sentence is misleading. We will change it in the revised version:
"To address this issue, uninitialised historical CMIP5 runs are usually considered …".

P4,L32: Replace "the natural variability" with "natural variability". Why use linear detrending to isolate natural variability when you have just introduced an uninitialised ensemble that can be used to quantify the skill associated with external forcing?
A: The forecast skill of a decadal prediction system may origin from two different "processes": a realistic prediction of the long-term trend and a suitable forecast of peaks on inter-annual timescales due to natural variability. To isolate the forecast skill for anomalies on inter-annual time scales, we originally detrended **all** datasets used in this study (as stated in the first paragraph of section 3.1), i.e. not only the hindcasts and the observations, but also the uninitialized historical runs. However, we agree with the Reviewer that detrending is not necessary when we use an uninitialized ensemble that allows us to determine the skill improvement relative to the externally-forced signal (see Reviewer's major comments). Hence, we decided to redo most of the analysis without detrending in the revised version (see also our answer to main comments). We will further replace "the natural variability" with "natural variability".

P5,L14: I think you mean "post-processed time series".
A: No, we mean pre-processed here, as they are processed before they are analysed by using different skill-metrics.

P5,L24: What is the basis for claiming that "skill should originate mainly from the initialization" as opposed to the external forcing? This has not been shown and shouldn't be assumed.

A: We agree with the Reviewer that this hypothesis is too speculative without analysing it in detail. We will therefore remove this clause in the revised version.

P5,L25-: What are F(y) and Fo(y)? Please explain the CRPS equation. What exactly is CDF and how is it computed?
A: The CRPS is defined as the quadratic measure of the discrepancy between the forecast cumulative density function (*F*) and the observed cumulative density function (*Fo)* of a variable *y*. The cumulative density function (CDF) of a real-valued variable *y* is defined as:
CDF(y) = P(y ≤ t),
where *P* is the probability that the variable *y* has a value of less than or equal to *t*.
We will add this information to the revised manuscript.

P6,L23-P7,L2: This is repetitive.
A: This paragraph will be removed in the revised manuscript.

Fig 2: Suggest using a nonlinear scale for MSESS, such as -9 to 0.9 as in Shaffrey et al. (2016, doi:10.1007/s00382-016-3075-x), because this metric is not symmetric about 0 in terms of relative improvements in MSE. Please clarify that these are for annual mean (ie, not seasonal mean) anomalies.
A: Following the Reviewers suggestion, we will use a nonlinear scale for Fig. 2 in the revised version and clarify that skillscores are for annual mean anomalies.

P7,L7: I presume the detrending has been performed similarly for observations and for the uninitialized historical runs? This isn't explicitly mentioned.
A: In the first paragraph of section 3.1 we stated that all datasets "are pre-processed in an analogous manner". However, to avoid misunderstanding we will add this information here again.

Table 1: What is the meaning of "The uninitialized historical ensemble has been used as reference dataset", given that this is a table of ACC scores? Am I correct that this table displays correlations computed from 5 data points (corresponding to the 5 start years)? Clearly the externally-forced trend is important and so this table should include ACC scores for the uninitialized historical runs for comparison.
A: We thank the Reviewer for this note. Up to now, no information of the uninitialized historical ensemble is included in Table 1. In the revised version we will include ACC scores for the uninitialized runs following the Reviewer's suggestion. As we use multi-annual means, correlations are actually computed from only 5 data points, which may be problematic as already criticized by the Reviewer. However, as shown in the responses to the main comments and in the attached figures, using 5 starting dates instead of yearly initialized hindcasts has mainly quantitative effects.

P7,L11: What is the meaning of "increases" ˘Ťrelative to uninitialized or relative to detrended?
A: The MSESS for the datasets with trend increases relative to the detrended time series. We will clarify this in the revised manuscript.

P7,L34: I would say Figure 2 shows more than a "slight shift".
A: This is indeed a too strong generalization of the results. Discrepancies between the two hindcast generations are rather small for temperature, but can be quite large for precipitation and wind speed, depending on the region. We will clarify this in the revised version.

P8,L2: Here and elsewhere delete "exemplary" as it is not being used properly.
A: Following the Reviewer's suggestion we will delete exemplary.

P8,L4: It's curious that Fig 2e agrees so well with Fig 3a, but Fig 2f is so different from Fig 3b. Can you offer any explanation? In my mind, it calls into question the significance of skill scores computed from 5 data points.
A: It is difficult to find an explanation for this issue. Aside from statistical reasons this might also be related to the detrending of the data. However, as we intend to replace this Figure by new ones obtained from calculations without detrending (see also response to main concerns), there is no point to speculate at this point in time. We will carefully check our new results for such discrepancies.

P8,L25-29: This discussion begs the question of why you are doing any detrending at all (see comment above)? The purpose of the uninitialized ensemble is precisely to allow you to discriminate between greenhouse-gas induced variability (including trends) and natural variability (including AMO-related trends). Detrending is confusing matter săĂ̆ Tjust compared initialized to uninitialized skill.
A: Again, we agree with the Reviewer in this point and will redo most of the calculations without detrending.

P9,L8: Change "whereas" to "and".
A: Will be changed.

P9,L9-11: This incomprehensible sentence needs a rewrite.
A: We will rephrase this sentence in the revised manuscript as it is indeed incomprehensible.

P11,L5: I don't understand this sentence.
A: If we would de-bias the CRPSS this would imply a different processing of the analysed datasets compared to the MSESS and the ACC and would make it difficult to compare the skill analysis. As stated in the introduction and in the response to the 4th specific comment of the Reviewer, this is exactly what we intend to avoid in our study. We therefore decided not to use a de-biased version of the CRPSS. However, as this is obviously stated incomprehensible, we will rephrase this sentence in the revised version.

P11,L24-28: This is because you are doing bootstrapping without replacement; if you allow replacement, then the spread does not necessarily diminish with ensemble size.
A: As stated in line 2 of page 11 of our manuscript our aim was not to do a bootstrapping, but to do permutations over all useful ensemble combinations. In our opinion the individual n-member ensembles should contain each ensemble member only once. Otherwise, the 10 member ensemble may in an extreme case consist of 10 times the same member, which in our opinion makes no sense. However, it is correct that the permutation without replacement results in a decline of the spread with increasing number of members. We will add this information to the paragraph in the revised manuscript.

[Figure]

Fig A1: (a,b) **Temperature correlation MPI-ESM-LR baseline1,** 10 members, lead-times year 1-5, observation: E-OBS. (c,d) Correlation baseline1 minus MPI-ESM-LR historical.

[Figure]

Fig A2: (a,b) **Precipitation correlation MPI-ESM-LR baseline1,** 10 members lead-times year 1-5, observation: E-OBS. (c,d) Correlation baseline1 minus MPI-ESM-LR historical.

[Figure]

Fig A3: (a,b) **Surface wind correlation MPI-ESM-LR baseline1,** 10 members lead-times year 1-5, observation: CCLM ERA 0.22°. (c,d) Correlation baseline1 minus MPI-ESM-LR historical.

---

## Author Response (AR1)

**Reviewer 1**

We thank the Reviewer for his/her thoughtful review and the specific suggestions. As one major point the Reviewer suggested to also show metrics that use the climatology as reference. We agree with the Reviewer that this would enhance the significance of our study. We therefore decided to include a new Figure 3 in the revised manuscript showing the MSESS as in Figure 2 but with the climatology as reference instead of the uninitialised historicals (see also answer to specific comments below). Please also note that we have repeated all calculations without detrending the time series following the suggestions of Reviewer 2. Therefore, not only all Figures have changed in the revised version, but we also had to revise large parts of the main text. Further, as we now use solely time series including the trend we have removed Table 1 and Table 2 in the revised manuscript, which showed the comparison between skill scores as derived from time series with and without trend. Instead, we added a section 4.4 and a new Figure 7 dealing with the effect of the low number of starting dates used in our study on the robustness of our results, again following the suggestions of Reviewer 2. All changes are marked in red in the revised manuscript.
Below point-to-point responses of the authors to all major and specific comments of the Reviewer are given, also in red.

**General comments**

This is a very good study that focuses on the potential merits of regional downscaling decadal climate predictions over Europe. Specifically, the MiKlip prediction system studied uses the low resolution MPI global decadal hindcast ensemble at T63 resolution and dynamically downscales these hindcasts over Europe using the COSMO-CLM model at 0.22_ horizontal resolution. Two 10 member ensemble regional hindcasts of 5 start dates are examined and verified against observational analyses of surface temperature, precipitation and low level wind using three different skill metrics, MSESS, CRPSS and ACC. The authors examine these metrics to answer the following questions: is there potential for skillful regional predictions in Europe? Does regional downscaling provide added value? and How does the skill of these predictions depend on ensemble size? The first two questions are answered affirmatively and for the last question ensemble size stabilizes the skill metrics MSESS and CRPSS at ten members but ACC skill depends on ensemble size beyond ten members. The manuscript meets all the criteria for publication and needs only minor changes.

**Specific comments:**

The manuscript could be improved in two ways that would increase the significance of the work. First, although there are significantly large regions in Europe where the skill of the initialized hindcasts is positive, there is also a large region in central Europe where the skill is negative. This is particularly true of the MSESS of temperature. Since the reference is forecast is an uninitialized ensemble of 20th century

simulations this raises the question as to the reason for this negative skill. The answer or some speculation to how it arises should be included in the article. In a similar vein, the authors do not include in their discussion any metrics that use the observed climatological distribution as the reference forecast, so that skill is measured solely using comparison with observations.

Answer: We thank the Reviewer for these helpful comments, which helped to improve our manuscript. Also in the revised Figure 2 the strong negative MSESS over Central Europe is still visible (without detrending the time series, see main comments of Reviewer 2). It is difficult (and also beyond the scope of our study) to find a physical interpretation for this negative skill. However, in order to get a

10 deeper insight to this issue, we have analysed the time series of spatial mean temperature over Prudence 4 (Mid-Europe) for the CCLM hindcasts, the historicals, and the observations (in this case E-OBS). The observed temperature over Mid-Europe strongly increases from dec1960 to dec1970. At the same time CCLM shows a strong decrease, so that it is out of phase of the observations during the first half of the considered period, while the historicals are closer to the observations for this time range. As

15 a consequence, the MSESS using the historicals as reference is strongly negative (as depicted in Fig. 2), although from dec1980 on the temperature curve of CCLM_b1 agrees well to the observations. We decided not to include an extra Figure with respect to this issue, but added a short paragraph in the revised manuscript.
Further, we followed the second suggestion of the Reviewer and performed an additional analysis

20 including the climatology as reference. The new Figure 3 in the revised paper shows the MSESS as in Figure 2, but with the climatology as reference (instead of the uninitialized historicals). The results are in this case "threefold": While for precipitation results look similar in Fig. 2 and Fig, 3, MSESS skill scores for temperature increase and for wind speed mainly decrease when using the climatology instead of the uninitialised historicals. We added a brief discussion on these results in the revised

25 version.

Technical corrections
Pg 2 Yaeger et al should be Yeager et al
30 A: Has been corrected throughout the manuscript.

Pg 8 stronger scattered should be more strongly scattered
A. We have changed it accordingly.

We thank the Reviewer for his/her thoughtful review and the specific suggestions. The Reviewer queried two major points: (i) the low number of starting dates used in our study (which has been discussed in detail in the original manuscript) and (ii) the detrending of the time series. As we generally agree with the Reviewer in both points and take the concerns seriously, we decided to include new Figures in the revised manuscript with respect to point (i) (see also answer to main concerns below). Further, we repeated all calculations without detrending the time series as suggested by the Reviewer and replaced the respective Figures in the revised manuscript. As the new calculations impact the majority of our results, a reformulation of most parts of the manuscript is necessary. These changes are marked in red in the revised manuscript. Note, that we have included a new Figure 3 in the revised manuscript. Following the suggestions of Reviewer 1 this Figure shows the MSESS as in Figure 2 but with the climatology as reference instead of the un-initialised historicals. Further, as we now use solely time series including the trend we have removed Table 1 and Table 2 in the revised manuscript, which showed the comparison between skill scores as derived from time series with and without trend.
The point-to-point responses of the authors to all major and specific comments of the Reviewer are also given in red in the following.

**Anonymous Referee #2**

*The paper gives a preliminary assessment of regional decadal prediction skill over Europe based on a high-resolution regional model forced with boundary conditions obtained from the low-resolution, global MiKlip prediction system. I deem the analysis preliminary because the "development and prospects" of the downscaling system are being assessed at a rather early stage when only 5 hindcast start dates have been completed using the regional model. This is a serious shortcoming that calls into question the reliability of skill scores (computed from 5 data pairs) that are used throughout to make statements about the benefits of downscaling for various fields in various European regions.*

*Two-tier decadal prediction involving regional downscaling is certainly a topic of high interest, but this manuscript has the feel of an internal technical note that documents some preliminary and very mixed results that are still clouded in uncertainty given the limited temporal sampling. Unless it can be shown (perhaps using the MPI baseline systems) that 5 start dates are sufficient to get an accurate estimate for the skill scores and fields of interest, then what is the point of all this? I suspect that 5 start dates is not sufficient, and that the skill scores reported here are very "noisy" as a result. This may contribute to*

*the mixed results and lack of strong take-away messages from this paper. It may be better to wait until more downscaled start dates have been completed before resubmission of this analysis.*

Answer:

We agree that the consideration of only five starting years lead to "noisier" results. But we argue, that this noisiness affects mainly chapter 4.1, which deals with the skill distribution over Europe. Chapters 4.2 and 4.3, which focus on the added value and the ensemble size dependency respectively, provide robust results even when using two times 40 simulations. Therefore, the major parts of the results are not strongly affected by the sample size issue.

[Figure]

*Figure R1: Spatial distribution of the MSESS for the multi-annual mean anomalies of lead years 1-5 in MPI_b1 for (a,b) temperature, (c,d) precipitation, and (e,f) wind speed. For the left panels five start years (dec1960, dec1970, dec1980, dec1990, dec2000) have been used, while for the right panels all start years from dec1960 to dec2000 are taken into account.*

Nevertheless, we performed a comparative analysis (as suggested by the reviewer) of the skill estimates for the three variables addressed in the paper derived from a) starting years every 10 years (1960, 1970,..,2000) as in the original manuscript and b) annual starting dates (1960-2000) for the global 10 member ensemble with MPI-ESM-LR baseline1. Baseline1 is the only ensemble used in the paper which provides 10 members throughout the whole hindcast period. The results (see attached Figure R1 showing the MSESS and new Figure 7 in the revised manuscript for the correlation) show a general qualitative agreement, though of course not a quantitative one. As expected, a larger sample size provides smoother skill estimates, less noisy than with the smaller sample size. But in general the findings for most regions that showed hindcast skill in the original manuscript are still correct for the extended ensemble. We have included this additional analysis in the new section 4.4 in the revised paper to point out, how and where the smaller sample size affects the findings and that way putting them into perspective. As we show the MSESS already in Fig. 2 and Fig. 3 of the revised paper we decided to only show the correlation in Fig. 7 but to also discuss the results for the MSESS (see Fig. R1) in the main text.

*Another main concern is the use of detrending, which probably exacerbates the sampling issues (how well-defined is a trend computed from 5 data points?). There is no real need to detrend since you have an uninitialized ensemble that allows you to determine the skill improvement relative to the externally-forced signal (yes, pure ACC will be higher, but you can show delta(ACC), i.e. the change in ACC relative to the uninitialized ensemble).*
*The quality of the writing is decent, but not high, and there are numerous instances of poor English construction (some noted below). A thorough proofreading is in order if this is to be resubmitted.*

A: We thank the Reviewer for this helpful comment. We agree that detrending is not necessary when using an uninitialized ensemble as reference. We therefore decided to redo all calculations of our study without detrending of the time series and included the new results in the revised manuscript. The new results generally differ only slightly from the outcomes of the analysis with detrending (cf. Figures in the original version to Figures in the revised manuscript). Altogether, we found a slight improvement. This is the case for both the absolute skill scores of the regional hindcasts (e.g. Fig. 2) and the added value of downscaling (e.g. Fig. 5 and 6). Noticeable differences are e.g. found for wind in Eastern Europe (see Fig. 2e,f). We have changed the text according to the new data processing procedure and to the new figures throughout the text. Additionally, following the Reviewers suggestion, we have included delta(ACC) as a measure for the change in ACC of the hindcasts relative to the uninitialized ensemble in Fig. 5 and 6. Further, we have carefully proofread the revised version of the manuscript as suggested by the Reviewer.

Specific Comments and Questions:
P2,L8: Here and throughout: "Yaeger" should be "Yeager".
A: Citation changed throughout the revised version.

P2,L11: It's not clear what the point is of the "while few" construction. Are you contrasting the large number of studies focusing on global metrics with the relatively few studies focusing on storm tracks, etc? Please rewrite.
A: We agree with the Reviewer that this sentence is misleading. We have rephrased it in the revised version.

P2,L13: What is this an example of? Why cite Sutton and Hodson (2005) in a paragraph focused on initialized decadal prediction?

A: We have removed this sentence in the revised manuscript.

P3,L7-18: The motivation for the present work needs to be clarified, particularly since it is not at all clear how the present study differs from the closely related recent MiKlip studies that have just been cited (Kadow et al. 2016; Mieruch et al. 2014; Haas et al. 2016; Moemken et al. 2016).

A: As stated in line 4-6 on page 3, the closely related MiKlip studies are difficult to compare, as they use different skill metrics, pre-processing methods and downscaling approaches. The unique feature of our study is that we use the same methods/metrics not only for the regional but also for the global prediction system. This enables us to give a more general assessment of the prospects of the MiKlip system with respect to basic near surface variables, which affect human life most (temperature, precipitation, and wind speed). However, we agree that this sentence is not sufficient to emphasize our motivation. We have rephrased it in the revised version.

P3,L11: This question is poorly phrased. Do you mean "depend on" the trend or "derive from" the trend?

A: We agree that this question is misleading. We have removed the second part of the question as we only consider time series with trend in the revised version, following the Reviewers suggestion.

P3,L15: This is a repetitive rephrasing of the questions just covered.

A: This paragraph has been removed in the revised manuscript.

P3,L29: I don't understand how ocean temperature and salinity can be nudged towards NCEP/NOAA reanalysis, since the latter is an atmospheric reanalysis.

A: The reviewer is correct. In baseline0 the ocean salinity and temperature anomalies were derived from a simulation with the ocean model MPI-OM forced with the NCEP re-analysis. We have changed the sentence accordingly.

P4,L11: Not clear what is meant by "Analog to the global data"?

A: To avoid misinterpretation, we have changed the text:

"The experiment includes downscaled hindcasts for dec1960, dec1970, dec1980, dec1990, and dec2000, with ten members per decade (hereafter CCLM_b0 and CCLM_b1). The regional ensembles therefore consist of the same time series like the global ensembles MPI_b0 and MPI_b1."

P4,L14: Replace "are" with "is".

A: According to the next Reviewers comment (P4, L16-19) we have changed the paragraph.

P4,L16-19: You already introduced the ERA-driven CCLM simulation in the first line of this paragraph, so consolidate your sentences into one brief description.

A: We have consolidated the sentences in the revised version.

P4,L21: I don't understand what you mean by "uninitialized model simulations started

from historical CMIP5 runs". Do you mean downscaled simulations that can be considered "uninitialized" counterparts to CCLM_b0 and CCLM_b1? Do you mean "preindustrial CMIP5 runs"?

A: We agree that this sentence is misleading. We have changed it in the revised version:
5 "To address this issue, uninitialised historical CMIP5 runs are usually considered …".

P4,L32: Replace "the natural variability" with "natural variability". Why use linear detrending to isolate natural variability when you have just introduced an uninitialised ensemble that can be used to quantify the skill associated with external forcing?

10 A: The forecast skill of a decadal prediction system may origin from two different "processes": a realistic prediction of the long-term trend and a suitable forecast of peaks on inter-annual timescales due to natural variability. To isolate the forecast skill for anomalies on inter-annual time scales, we originally detrended **all** datasets used in this study (as stated in the first paragraph of section 3.1), i.e. not only the hindcasts and the observations, but also the uninitialized historical runs. However, we
15 agree with the Reviewer that detrending is not necessary when we use an uninitialized ensemble that allows us to determine the skill improvement relative to the externally-forced signal (see Reviewer's major comments). Hence, we decided to redo all the analysis without detrending in the revised version (see also our answer to main comments). As a consequence we have removed this paragraph in the revised manuscript.

P5,L14: I think you mean "post-processed time series".
A: No, we mean pre-processed here, as they are processed before they are analysed by using different skill-metrics.

25 P5,L24: What is the basis for claiming that "skill should originate mainly from the initialization" as opposed to the external forcing? This has not been shown and shouldn't be assumed.

A: We agree with the Reviewer that this hypothesis is too speculative without analysing it in detail. We therefore have removed this clause in the revised version.

P5,L25-: What are F(y) and Fo(y)? Please explain the CRPS equation. What exactly is CDF and how is it computed?
A: The CRPS is defined as the quadratic measure of the discrepancy between the forecast cumulative density function (*F)* and the observed cumulative density function (*Fo)* of a variable *y*. The cumulative
35 density function (CDF) of a real-valued variable *y* is defined as:
CDF(y) = P(y ≤ t),
where *P* is the probability that the variable *y* has a value of less than or equal to *t*.
We added this information to the revised manuscript.

40 P6,L23-P7,L2: This is repetitive.
A: This paragraph has been removed in the revised manuscript.

Fig 2: Suggest using a nonlinear scale for MSESS, such as -9 to 0.9 as in Shaffrey et al. (2016, doi:10.1007/s00382-016-3075-x), because this metric is not symmetric

about 0 in terms of relative improvements in MSE. Please clarify that these are for
annual mean (ie, not seasonal mean) anomalies.
A: Following the Reviewers suggestion, we have used a nonlinear scale for Fig. 2 and the new Fig.3 in
the revised version and clarified that skill scores are for annual mean anomalies in the Figure captions.

P7,L7: I presume the detrending has been performed similarly for observations and for
the uninitialized historical runs? This isn't explicitly mentioned.
A: In the first paragraph of section 3.1 of the original manuscript we stated that all datasets "are pre-
10    processed in an analogous manner". However, as we decided to keep the trend in the datasets for our
analysis, we have removed this paragraph in the revised manuscript.

Table 1: What is the meaning of "The uninitialized historical ensemble has been used
as reference dataset", given that this is a table of ACC scores? Am I correct that this
15    table displays correlations computed from 5 data points (corresponding to the 5 start
years)? Clearly the externally-forced trend is important and so this table should include
ACC scores for the uninitialized historical runs for comparison.
A: We thank the Reviewer for this note, which is correct. However, as we keep the trend in the
analysed datasets in the revised manuscript as suggested by the Reviewer, we decided to remove
20    Table 1 and Table 2 in the updated version as they originally showed "trend versus detrended" results.
With respect to the correlation, we now included ACC scores for the uninitialized historical runs in
Figures showing correlation scores.

P7,L11: What is the meaning of "increases" ̆Trelative to uninitialized or relative to
25    detrended?
A: The MSESS for the datasets with trend increases relative to the detrended time series. However,
this paragraph has been removed in the revised version (see answer to comment above).

P7,L34: I would say Figure 2 shows more than a "slight shift".
30    A: This is indeed a too strong generalization of the results. Discrepancies between the two hindcast
generations are rather small for temperature, but can be quite large for precipitation and wind speed,
depending on the region. We have clarified this in the revised version.

P8,L2: Here and elsewhere delete "exemplary" as it is not being used properly.
35    A: Following the Reviewer's suggestion we have deleted exemplary throughout the manuscript.

P8,L4: It's curious that Fig 2e agrees so well with Fig 3a, but Fig 2f is so different
from Fig 3b. Can you offer any explanation? In my mind, it calls into question the
significance of skill scores computed from 5 data points.
40    A: It is difficult to find an explanation for this issue. Aside from statistical reasons this might also be
related to the detrending of the data. However, in the revised version we included a new Fig. 3 showing
the MSESS with the climatology as reference.

P8,L25-29: This discussion begs the question of why you are doing any detrending at
45    all (see comment above)? The purpose of the uninitialized ensemble is precisely to allow
you to discriminate between greenhouse-gas induced variability (including trends)

and natural variability (including AMO-related trends). Detrending is confusing matter sẵ Tjust compared initialized to uninitialized skill.
A: Again, we agree with the Reviewer in this point and have repeated all calculations without detrending. Therefore, this paragraph has been removed in the revised manuscript.

P9,L8: Change "whereas" to "and".
A: Has been changed.

P9,L9-11: This incomprehensible sentence needs a rewrite.
A: We have rephrased this sentence in the revised version.

P11,L5: I don't understand this sentence.
A: If we would de-bias the CRPSS this would imply a different processing of the analysed datasets compared to the MSESS and the ACC and would make it difficult to compare the skill analysis. As stated in the introduction and in the response to the 4[th] specific comment of the Reviewer, this is exactly what we intend to avoid in our study. We therefore decided not to use a de-biased version of the CRPSS. However, as this is obviously stated incomprehensible, we have rephrased this sentence in the revised version.

P11,L24-28: This is because you are doing bootstrapping without replacement; if you allow replacement, then the spread does not necessarily diminish with ensemble size.
A: As already stated in the original manuscript our aim was not to do a bootstrapping, but to do permutations over all useful ensemble combinations. In our opinion the individual n-member ensembles should contain each ensemble member only once. Otherwise, the 10 member ensemble may in an extreme case consist of 10 times the same member, which in our opinion makes no sense for decadal prediction purposes. We would therefore keep the method without replacement in the revised version depending on the Editors decision. However, it is correct that the permutation without replacement results in a decline of the spread with increasing number of members. We have added this information to the paragraph in the revised manuscript.

[revised manuscript text omitted]

---

## Referee Report (RR1)

In the field of multi annual prediction this paper reflects the state of the art and is the first to demonstrate skill on the regional scale. This manuscript should be published nearly as is. I have only some minor suggested editorial corrections.

Editorial changes:
In section 2 on page 3 the authors should note that anomaly initialization is being used in all ensemble members, global and regional. This is only stated on page 14 and the lack of bias or drift in forecasts is puzzling until then.

In the discussion of the MSE score at the top of page 6, n and N should be defined.

On page 12 line 20 'chapter 4.1' should be 'section 4.1'

Suggest that 'With is respect' be changed to 'With respect to this' on page 13 line 27

---

## Editor Decision (ED1)

Formulas on page 5 and 6 are difficult to read. Would you please make corrections on page 5 and page 6 to have formulas more consistent with the definitions.

**page 5, line 25: change to**

$$MSESS(H, R, O) = 1 - \frac{MSE_{hind}}{MSE_{ref}}$$

**page 5, line 27: change to**

$$MSE_{hind} = \frac{1}{N}\sum_{i=1}^{N}(\overline{H_i} - O_i)^2 \text{ and } MSE_{ref} = \frac{1}{N}\sum_{i=1}^{N}(\overline{R_i} - O_i)^2$$

**page 5, line 29-30: change to**

... downscaled hindcasts $(H_i)$ and the verification data $(O_i)$, and $MSE_{ref}$ is the mean squared error of a reference dataset $(R_i)$.

**page 6, line 1-2 add:** A MSESS could be between minus infinity and one with positive values meaning that hindcasts are close ...

**and remove (-infinity,1] on line 10 and line 12**

**page 6, line 3: change to** ".., the MSESS with the climatology as reference (i.e. $R_i \equiv \bar{O}$ ) can be decomposed as follows"

**page 6, line 5: change to**

$$MSESS(H, \bar{O}, O) = r_{H,O}^2 - \left(r_{H,O} - \frac{S_H}{S_O}\right)^2$$

**page 6, line 6: give direct definition of correlation $r_{H,O}$ between Hi and Oi**

**page 6, line 6-8: change X→ H**

**page 6, line 10: change indices –** HO to H,O; RO to R,O; H,R.O to H,R,O

**page 6, line 12: change to**

$$MSESS(H, R, O) = \frac{MSESS(H, \bar{O}, O) - MSESS(R, \bar{O}, O)}{1 - MSESS(R, \bar{O}, O)}$$

**page 6, line 24: change $\sigma_H, \sigma_O$ to $S_H, S_O$**

---

## Author Response (AR2)

**Point-to-point replies to Reviewer comments for manuscript esd-2017-70**

Comment by the authors:

Please find below point-to-point replies to the comments of anonymous Referee #1 and #3. Responses are given in red. Changes in the revised manuscript according to the comments are also given in red. Note that we have replaced old Fig. 2-7 in the revised version according the suggestions of Referee #3. Further, old Fig. 5 is now split into three Tables (Table 1-3 in the revised manuscript). New Figures 2-4 and 6 are attached at the end of this document as Figures R1-R4.

Anonymous Referee #1

In the field of multi annual prediction this paper reflects the state of the art and is the first to demonstrate skill on the regional scale. This manuscript should be published nearly as is. I have only some minor suggested editorial corrections.

Editorial changes:

In section 2 on page 3 the authors should note that anomaly initialization is being used in all ensemble members, global and regional. This is only stated on page 14 and the lack of bias or drift in forecasts is puzzling until then.

Answer: As suggested by the Reviewer we noted that anomaly initialisation is used for all ensemble members in section 2 of the revised manuscript.

In the discussion of the MSE score at the top of page 6, n and N should be defined.

Answer: We thank the Reviewer for this hint. We have clarified this in the revised manuscript.

On page 12 line 20 'chapter 4.1' should be 'section 4.1'

Answer: We have changed „chapter" to „section" in the revised version.

Suggest that 'With is respect' be changed to 'With respect to this' on page 13 line 27

Answer: As suggested by the Reviewer, we have reformulated this sentence.

Anonymous Referee #3

**General Comment:**
The key issues raised by reviewer 3 below concern the quantification of the significance of the results and in particular the quantification of uncertainty. However, we think that these doubts are mainly due to the presentation of the results but not due to the results per se. We have addressed these issues throughout the whole revised manuscript:

- By including significances/uncertainties/quantifications using a bootstrap approach in the revised manuscript (see attached new figures R1-R4, replacing old figures 2, 3, 4 and 7, and specific comments below), we demonstrate that the prediction skill may indeed be significantly improved by dynamical downscaling for the selected meteorological variables in certain regions.
- By including additional quantitative arguments and significances using a t-test

**Recommendation to the editor**

**Suggestions for revision or reasons for rejection (will be published if the paper is accepted for final publication)**
The manuscript "Development and prospects of the regional MiKlip decadal prediction system over Europe: Predictive skill, added value of regionalization and ensemble size dependency" by Mark Reyers et al. provides an insight into regional decadal climate prediction activities that have been undertaken within the MiKlip initiative.

The manuscript targets at answering four central research questions:
a) Is there a potential for skillful regional decadal predictions in Europe?
b) Does the regional downscaling provide an added value for decadal predictions?
c) Does the regional decadal predictive skill depend on the ensemble size?
d) How does the sample size affect the skill estimates?

I was not involved in the first round of the review process. I read the manuscript first without having a look at the response letter before to remain unbiased in this respect. I appreciate the effort the authors spent so far to develop the manuscript. However, even though a regional decadal prediction system is something of great interest for the climate science community, I have to suggest rejecting the manuscript given its current shape and scientific focus.
This for a number of general reasons:

1. The authors state that MiKlip is the first effort worldwide establishing a decadal prediction system for the regional scale (I guess they mean the included dynamic downscaling here.). To my knowledge this statement is absolutely correct but this is for a reason. The dynamical downscaling of the decadal predictions is associated with large computational costs. The authors have to prove that these costs are outperformed by benefits in prediction skill for certain variables or phenomena. The manuscript currently fails completely to deliver this evidence:
Answer: We agree with the Reviewer that dynamical downscaling of a large ensemble of decadal hindcasts has high computational costs. However, the aim of this study is to evaluate the dynamical downscaled hindcasts, which is a first step to establish a regional prediction system. Once the regional system is evaluated, only the decadal predictions need to be

downscaled, and not the whole ensemble again. As these predictions will consist of one start year in an operational system, the computational costs are acceptable. There are further justifications for the choice of the dynamical downscaling approach, which are given in our answers to the Reviewer's comments below.

1.1. I do not understand why the authors chose perennial means of temperature, precipitation, and wind as the basis for their validation of a regional decadal prediction system. The variability of such multi-year means of primary meteorological parameters will be mainly determined by low-frequency variability of large-scale circulation patterns (e.g. NAO).

Answer: Multi-year means are used in most studies dealing with the validation of decadal prediction systems (Doblas-Reyes et al., 2013; Marotzke et al., 2016, Meehl et al., 2014 and references therein). Their usage is even recommended by Goddard et al. (2013) and Boer et al. (2016). Therefore, the use of multi-year means in the analysis of decadal predictions is a state-of-the-art approach. The Reviewer is correct in the sense that the low-frequency determines the multi-year variability of the primary meteorological parameters. This is in fact the main source of decadal predictability. The leading mode of variability on this time-scale is the Atlantic Multidecadal Variability (AMO, Smith al al., 2012; Sutton and Hodson, 2005 etc), which has a strong impact on the European climate. We have added a short paragraph in the introduction section of the revised manuscript.

*Smith D. M., Scaife A. A., and Kirtman B. P. (2012) What is the current state of scientific knowledge with regard to seasonal and decadal forecasting? Environ. Res. Lett., 7(015602), doi: 10.1088/1748-9326/7/1/015602*

There is no need to employ an RCM to model this, this is done (more or less successfully) by the driving GCM. The only benefit provided by the RCM for these variables' means is (hopefully) some correction of the conditional bias (improved variance) due to a better resolution of orography and land-use. However, such (conditional) bias-correction could be done much cheaper by means of statistical downscaling. I would like to read an argumentation why the authors expect regional downscaling to be valuable in this respect.

Answer: We think that dynamical downscaling has many advantages compared to statistical downscaling. First, all variables are physically consistent in dynamical downscaled model runs, which is not the case for statistical downscaling of multiple variables. Second, a statistical method has to be developed for all variables of interest separately, which is associated with high technical efforts. And third, in the statistical downscaling the relationship between large-scale and regional variables is only known for the period for which the method is developed, but it is unclear if it remains the same in future decades. This issue is overcome by the dynamical downscaling due to its physical consistency. Further, and as mentioned above, the aim of this study is to evaluate the regional system towards a later operationalisation. In future applications, the dynamically downscaled ensemble will be used to analyse user-oriented parameters (e.g. wind gusts, heavy precipitation events potentially causing floods, growth degree days, …), which often consist of a combination of different variables on the regional scale. Hence, ensembles of numerous consistent variables are required for this purpose, which can only be delivered by a dynamical downscaling approach. Finally, the regional decadal hindcasts may be used for impact modelling or as forcing data for hydrological models, which requires physically

consistent data. We have added a brief discussion of the advantages of dynamical downscaling in the introduction section of the revised manuscript.

1.2 Additionally, I have problems interpreting/understanding the "proof" of added value (i.e. skill) the downscaling yields compared to the GCM predictions. I understand that the authors present Fig. 4 & 5 targeting this question. The problem with Fig. 4 is that the key message (RCM prediction better than GCM prediction) is based on qualitative evaluation. I agree that it seems from Fig. 4 that skill scores are shifted up and right in the scatter diagrams when comparing the RCM predictions to the GCM predictions. But this is just a result from eye-balling the plots. There is no direct quantification of the gain (and its significance). In principle it would be possible to show differences between 4a&b, 4c&d, and 4e&f, respectively. However, I do not really suggest to go for this solution (calculating the differences of these scatter diagrams). I would rather propose to present skill maps (as done for Fig 2 & 3) again, that is a map plot showing skill of RCM-predictions with the GCM-predictions as reference forecast.

Answer: We thank the Reviewer for this comment, which includes helpful suggestions. Following the Reviewer's suggestion, we have replaced the old Fig. 4 by a map plot showing the MSESS of the RCM-predictions using the MPI predictions as reference dataset. This enabled us to quantify where the dynamical downscaling significantly improves the prediction skill. We have changed the text accordingly in section 4.2 of the revised manuscript.

Fig.5 is problematic, too. First, I do not understand what the authors indicate by the background color. They say it is the "absolute skill" (page 18 line 32). Is this supposed to mean that the background color indicates whether there is skill over the climatological forecast, while the dots mark skill over the GCM-predictions? If so, that is fine but should be explicitly written in the manuscript (and the figure caption). And I do not understand the difference between ACC and delta_ACCin this diagram. An additional problem of this plot (as with most of the analyses) is that it does not take the uncertainty into account (see my dedicated issue below). When do the authors consider skill scores to be equal??? Strictly speaking, every prediction should be considered equal that is not proven to be significantly better than the reference forecast. My assumption is, that this would lead to a totally yellow table diagram in Fig. 5.

Answer: We agree that Fig. 5 is a very busy graphic. For the revised manuscript, we split the information into three tables, including quantitative numbers and use the t-test to derive a measure where the skill differences are significant. Additionally, we decided to remove delta_ACC in this section and include results from the Murphy decomposition instead (see also last comment).

My general doubt regarding the benefit from dynamical downscaling is further supported by Fig. 6. Even though compiled for another research question (the ensemble size-dependent bias), this figure shows impressively (for the example of the IP-region) that the downscaling hardly yields any benefit. If the authors would quantify the uncertainties, I am pretty sure that all RCM-curves would be situated within the confidence intervals of the respective GCM curves.

Answer: The box-whisker plots shown in old Fig. 6b represent the ensemble spread resulting from the choice of the ensemble members. As this spread represents a part of the uncertainty, we agree with the Reviewer that no significant differentiation of the skill scores can be determined for small ensemble sizes. However, for an ensemble size of seven and more, the

confidence interval strongly decreases. We have now added box-whiskers in new Fig. 5d-5f in the revised manuscript for different purposes (see revised main text). In particular from new Fig. 5f it is visible that for the ensembles of more than seven members the CCLM_b0 curve is well above the confidence interval of MPI_b0. Hence, as already shown in the new Fig. 4 (see also answer to comment 1.2 above), the prediction skill may indeed be significantly be improved when downscaling the full ensemble of ten members depending on the variable and the region.

All this sums up to my point of view that the manuscript fails to provide evidence of (significantly) added value achieved by downscaling for the analyzed variables. From my point of view, the great potential of employing a non-hydrostatic RCM for decadal predictions could be in the representation of extremes, especially for precipitation. However, frequencies of extremes or something similar are not addressed by the manuscript.
Answer: Regarding the Reviewer's criticism given in comment 1.1 and 1.2, we can understand the doubts of the Reviewer regarding the added value of downscaling. However, we think that these doubts were mainly due to presentation of the results, and not due to the results per se. By including significances/uncertainties/quantifications in the revised manuscript (see our answers above), we could demonstrate that the prediction skill may indeed be significantly improved by dynamical downscaling.
Again, the aim of our study is to evaluate the regional ensemble. Extremes and other user-relevant parameters will be examined in future applications.

2. A point already issued by one of the last round's reviewers (and explicitely mentioned in the editor's decision) is the low number (5) of initializations. The authors present figure 7 in the revised manuscript and state that these plots prove a general compliance of the results derived from five initializations with those based on 41 initializations. This is true for temperature where skill basically exists for whole Europe due to extrenally forced global warming. However, for precipitation and wind there are some important differences between results based on only 5 and 41 initializations, respectively. I resist from naming them explicitely, the key message here is that the uncertainty associated with all skill scores calculated for the manuscript is huge. And of course the skill estimates based on 41 initializations in Fig. 7 still feature substantial uncertainty, they are not the ultimate "true" skill. This is the essential problem of the manuscript: uncertainty is never quantified. Without such quantification of uncertainty/significance, the results cannot be taken scientifically serious.
Hence, the manuscript is not able to achieve its central aim that is providing evidence or at least significant supportive indication of a potential benefit from dynamical downscaling for decadal climate prediction.
Answer: The Reviewer is correct that a low number of hindcast starting dates poses a serious problem in the analysis of decadal predictions in general (and in this study in particular). This was recognized after CMIP5 (Meehl et al., 2013) and changed for CMIP6 (Boer et al., 2016). As already stated in the conclusions of the paper, a result of the efforts presented in this manuscript is to use annual starting dates on the regional scale for further hindcast generations. New Figure 6 and research question d) explicitly address this issue. We think, that this part is important for the paper (and was requested by the other reviewers), i.e, to show how far the results obtained for the other research questions can be transferred to an ensemble with a larger sample size. To address the concern raised by the Reviewer regarding the estimation of the uncertainty, we have now included the significance at the 95%-level of the resulting skill scores derived from a bootstrapping approach (see Fig. R4 and others). The results provide evidence of the statements made in our paper, that there are regions which

provide a significant skill. Fig. R4 is included as new Fig. 6 in the revised manuscript and we have adapted the text accordingly.

3. The scientific questions c) and d), both targeting the ensemble-size-dependent bias of (regional) prediction skill are not worth tackling in a way done by this manuscript. From my point of view, the manuscript does not add anything scientifically new to the literature in this respect. Of course, it is absolutely important to account for this skill-bias when comparing predictions made by ensembles of differing size and it might be helpful to indicate this bias even for analyses of ensembles with same size in order to quantify potential skill of (hypothically) larger ensembles. But there is no doubt about the general question if there is a bias: there is! The authors should have a look into original papers addressing this issue, such as Murphy (1990), Ferro (2007), and Sienz et al. (2016; there might by older papers than Sienz et al., tackling the bias of MSE/RMSE but I currently remember only this rather recent one). They will learn that the skill bias for all metrics used is essentially related to the variance of the target parameter and the actual bias is a result from the additional noise component due to an inaccurate estimate of the signal due to the limited ensemble size. The theoretical behaviour of the bias depending on ensemble-size is thus also known and can be estimated with existing approaches given by these papers. By the way, this disproves the author's statement that only the CRPSS could be corrected, which is why they refrained from using a de-biased version of the CRPSS in order to maintain comparability to the other metrics (page 20 lines 26-30).

Murphy, J.M. (1990): Assessment of the practical utility of extended range ensemble forecasts. Q.J.R. Meteorol. Soc., 116: 89-125. doi:10.1002/qj.49711649105
Ferro, C.A.T. (2007): Comparing Probabilistic forecasting systems with the brier score. Wea. Forecasting, 22: 1076-1088. doi:10.1175/WAF1034.1
Sienz, F., Müller, W.A., Pohlmann, H (2016): Ensemble size impact on the decadal predictive skill assessment. Meteorol. Z., 25(6): 645-655. doi:10.1127/metz/2016/0670

Answer: From our point of view, the ensemble size (e.g. number of realizations) and the sample size (which takes the number of starting dates and respective climate conditions into account) pose different research questions. This is in agreement with the findings by Sienz et al. (2016). The previous studies mentioned the use of either synthetic data (Sienz et al., 2016) or data from weather prediction. In this manuscript, the intention is to show the effect of increasing the ensemble size in actual (regional and global) decadal predictions. It was not the intention here to estimate the limit for an infinite ensemble size. The current decadal prediction systems rarely use large ensemble sizes. For instance, the CMIP5 recommendations proposed an ensemble size of three members for each starting year (Taylor et al., 2012). In new Figure 5, we show that such a low ensemble size of three members is clearly insufficient. Furthermore, the results indicate that the minimum required number of members depends both on the meteorological parameter and on the chosen metric. We agree that the variability between the sub-ensembles for very small numbers of members makes the results indistinguishable – as shown in new figures 5d-f. But we could also provide evidence that with a higher number of members the results for some parameters and metric become distinguishable. We have included box-whisker plots for additional ensembles and variables in the new Figures 5d-f, in order to allow a better quantification of the needed minimum ensemble size. In the revised version of the manuscript, we have improved the explanation.

4. The authors present their analyses based on three metrics, that is the MSESS, the ACC, and the CRPSS. However, there is hardly any information on why the authors chose these three metrics: What is the additional information that is provided by ACC and CRPSS

compared to using the MSESS only? What do we learn from differences between the various metrcis presented in Fig. 5 (e.g. What does it mean that CRPSS for the FR-region wrt wind is positive for the RCM-prediction compared to the GCM-prediction but negative for all other skill metrics?). There are a few hints in Sec. 3.2 when introducing the metrics, however, some of them are rather misleading. This is especially true for the CRPSS which does not only contain information about the "reliability" of forecasts but also about "resolution" (see page 15 line 16). Additionally, it does contain much more information than only checking whether the ensemble spread is an adequate representation of forecast uncertainty (compare page 20 lines 25-26). Last but not least, the authors do not consider the relationship between the different metrics. This is most relevent for the MSE(SS) and the ACC. Following Murphy (1988) the MSE for a forecast f measured against the obervation o can be decomposed into:

$$MSE(f,o)=(<f> - <o>)^2 + s\_f^2 + s\_o^2 - 2\ s\_{fs\_o}\ R\_{fo}$$

where $<f>$ and $<o>$ are the means of f and o, respectively, $s\_f^2$ and $s\_o^2$ are the sample variance of f and o, respectively, and $R\_{fo}$ is the sample correlation of f and o. As soon as anomalies are considered, as done in the study by Reyers et al., the first term becomes 0 for all instances. A positive MSESS (meaning a lower MSE than the reference forecast) hence can be achieved by a smaller forecast variance (this is where the ensemble size is relevant, by the way => larger ensembles yield smaller noise variance) or an increase in correlation. The latter is additionally calculated for the present study, although - as just described - closely linked to the MSE(SS). This might make sense if drawing the right conclusions, especially for cases where ACC and MSESS develop differently. However, currently it seems that the study just somehow "quantifies skill" (whatever that means) without any attempt to assess what the reasons are why the RCM is performing better/worse than the GCM predictions (and still failing because not quantifying the uncertainty). I suggest that the authors present less skill metrics (maximum of two) but clearly think about about which to choose and really spend some effort to disentangle the different meanings provided by these. One natural choice would be to choose MSESS and CRPSS which comprises a maximum of information. However, it might be better to stay with deterministic scores only. This would be the MSE(SS) as primary verification metric with the ACC as complementary metric in order to assess whether it is really the signal that is predicted better/worse or whether it is only a change in forecast variance (which might be the case when comparing RCM and GCM predictions).

Murphy, A.H. (1988): Skill Scores Based on the Mean Square Error and Their Relationships to the Correlation Coefficient. Mon. Wea. Rev., 116, 2417-2425, doi:10.1175/1520-0493(1988)116<2417:SSBOTM>2.0.CO;2

Answer: We thank the Reviewer for this helpful comment. Regarding the choice of metrics and their application, the authors follow the recommendations for the verification of decadal predictions as proposed by Goddard et al. (2013) as cited in the manuscript (page 5, lines 15f). We agree with the Reviewer that the differences and relations of the metrics should be explained in more detail. Following his/her suggestions, we have omitted the CRPSS in the revised version of the manuscript and focus only on MSESS and ACC. Further, we now interpret our results based on the Murphy decomposition and analyse in more detail the contributions of the correlation and the conditional bias to the MSESS skill. In addition, we have clarified where the downscaling is able to improve the skill, which will certainly improve the scientific validity of our study (see also answers to comments above). The Murphy decomposition is described in chapter 3.2 of the revised version.

[revised manuscript text omitted]

---

## Author Response (AR3)

**Point-to-point replies to Reviewer comments for manuscript esd-2017-70**

Comment by the authors:

Please find below point-to-point replies to the comments of the anonymous Referee. Responses are given in red. Changes in the revised manuscript according to the comments are also given in red.

General comments: ------------------

1. I suggest changing the order of results shown in section 4.1 (Fig. 2 & 3). It seems more logical to me - and hence easier to follow for the reader - to present the skill analyses starting with a reference forecast that would be expected to be rather easy to outperform (climatological forecast) before using reference forecasts that are a priori known to contain skill (the uninitialzed simulations). The results in section 4.2 then continue this logical order by using even more sophisticated reference forecasts, the initialized global predictions.

Answer: We have changed the order of the results in section 4.1 as suggested by the Reviewer. Hence, Fig. 2 now shows the MSESS using the climatology as reference, while in Fig. 3 the uninitialized historicals are used. We have changed the text in section 4.1 accordingly in the revised manuscript.

2. I have the feeling that you did not fully get the point of my comment 3 of last round's review, that is the issue of ensemble size and its impact on skill. I agree that this issue has to be considered seperately from the issue of few initializations (rather use this term than "sample size", from my understanding the "sample size" comprises both, the number of initializations times the ensemble size). Let's focus on the ensemble size issue here. I still insist on my point of view that this problem of an ensemble-size dependent bias is known (and solutions or let's say workarounds). This is a mathematical issue of the various skill metrics as they are and has nothing to do with the origin of the actual data (synthetic, weather or climate prediction, global or regional). In the end the bias itself and its behaviour with a growing number of ensemble members depends only on the signal-to-noise-ratio (which crucially depends on the variance of the targeted variable). Anyway, as I wrote in my previous review: I appreciate your effort of tackling this issue (many studies do not, and probably aren't even aware of this problem). However, your results do not add anything new to the scientific literature regarding the existence of this bias. That is why, I oppose against your research question 3 (Does [...] skill depend on ensemble size?). To resolve this issue, I suggest the following: Please reformulate your research question 3 to something like "How does ensemble size impact regional decadal prediction skill?" It's a small change but this "how" makes a difference from my point of view. And please, carefully revise your text in a sense that it does not sound anymore as if the basic question (dependence yes or no) is something that has never been addressed before. Maybe you can write something like "the ensemble-size dependent skill bias has never been demonstarted based on regional decadal climate predictions before".

Answer: Following the Reviewer's suggestion we have reformulated research question 3 in the revised manuscript. Further, we have revised the text in section 4.3 in the sense suggested by the Reviewer.

3. Regarding your research question 4: As shortly mentioned above, please refrain from naming this issue as a matter of sample size. The sample size in the end is determined by the ensemble size and the number of initializations. Please make clear that you address the issue of few initializations here!

Answer: We have changed this research question as suggested. Further, we have revised section 4.4 and parts of the discussion accordingly.

Specific comments: ------------------

1. Page 2, line 15: I suggest including a reference to Eade et al. (2012) related to the prediction of extremes.

Answer: We have included the reference in the revised version.

2. Page 2, lines 31-33: Please split in two sentences. First one to end after "...techniques". And I suggest to replace "outstanding" with "exceptionally".

Answer: We have reformulated this part as suggested by the Reviewer.

3. Page 3, line 7: "aspread" should be "spread".

Answer: It is now changed to "spread".

4. Page 3, line 12: Replace "for the decadal predictability" by "regarding skill". The reference to the decadal timespan comes later in the very same sentence. And "predictability" actually is to be differentiated from "prediction skill" but this is another discussion and unfortunately done inaccurate by many colleagues.

Answer: We have reformulated it in the revised manuscript.

5. Page 3, lines 21-22: Please reformulate reserach questions 3 and 4 according to the suggestions made in my general comments.

Answer: We have reformulated research question 3 and 4 as suggested by the Reviewer.

6. Page 3, line 32: As already mentioned by another reviewer during the discussion stage of this manuscript (and answered correctly by you), ocean temperature and salinity are NOT taken from NCEP/NOAA reanalysis. Please pay attention to this issue and describe the initialization procedure correctly.

Answer: We clarified the procedure in the revised manuscript: "The first generation (baseline0; Müller et al., 2012, Matei et al. 2012) is initialised with oceanic conditions from an experiment,

where surface fluxes from the NCEP/NOAA reanalysis (Kalnay et al., 1996) were assimilated into the ocean model MPI-OM. The anomalies of ocean temperature and salinity from this experiment were then used to initialize the decadal hindcasts in the coupled model."

7. Page 4, lines 7-9: I suggest removing the sentence regarding the downscaling from this paragraph. It's almost exactly repeated in the following paragraph (lines 15-17) which is specifically dedicated to the downscaling.

Answer: We agree with the Reviewer, that this is a repetition. However, we prefer not to remove the whole sentence, as it contains abbreviations which are used throughout the manuscript. Instead we have reformulated this sentence such that the downscaling is not mentioned anymore.

8. Just as a matter of curiosity: Did you test whether it makes a difference for your skill estimates if you take (interpolated) winds from ERA-reanalyses directly instead of using the ERA-driven CCLM-simulation as a reference?

Answer: This is an interesting point. However, we did not test this in our study. We can only speculate, but we assume that it may indeed impact the skill estimates when using winds directly from ERA-reanalysis, as some physical mechanisms are not captured when "simply" interpolating ERA-Interim winds to a finer grid.

9. Page 4, lines 27-28: It is not correct that "historical" simulations are forced ONLY by aerosol andgreenhouse gas concentrations. There is a number of other external forcings that are prescribed. Please rephrase accordingly.

Answer: That is of course correct. We rephrased the sentence in the revised manuscript: "With this aim, a 10-member ensemble of uninitialised MPI-ESM-LR historical runs started from a pre-industrial control simulation are used, which use observed natural and anthropogenic forcings (e.g. aerosol and greenhouse gas concentrations among others) for the period 1850-2005 (e.g. Müller et al., 2012)."

10. Page 4, lines 31-32: What is the interpolation method you used? Did you test for alternative interpolation methods and the related impact on skill? It's quite a range of resolutions you are using here. Given that you interpolate from much coarser but also from (slightly) higher resolution (featuring a rotated grid) to the 0.25deg-grid, I guess bi-linear interpolation for all datasets would be the most appropriate solution. In any case, please indicate your the method chosen by you.

Answer: We have actually used the bi-linear interpolation method in our study to interpolate all datasets to the E-OBS grid (0.25°x0.25° resolution). We have clarified this in the revised version. With respect to the second question, we have tested different interpolation methods in other studies, and found that the impact on the skill estimates are negligible. TODO: Please check my reformulation and comment in the manuscript.

11. Page 5, lines 1-2: How were the anomalies calculated for the hindcasts, that is to mean, how did you calculate a climatology from the hindcasts? This question may sound stupid, but the devil is in the detail and it might even be, that your hindcasts feature drifts (even though it is anomaly initialization). I think the latest recommendation in this respect is to define a baseline period that is covered by the same number of initializations for all lead times and then calculate a climatology for every single lead time separately. This is not possible for you given that you have initializations only five-yearly. I don't ask for a change in your approach here (whatever you did) but please describe precisely how you did it.

Answer: We did exactly what we have written in the manuscript. To calculate the anomalies, we removed the mean over the period 1961-2010 from the hindcasts and the observations, respectively. Tests using a different reference periods did not reveal changes in the results presented here.

12. Page 5, line 4: I suggest replacing "mainly" by "partly". You do spatial averaging over the regions only for the results presented in tables 1-3. However, the number of plots where you did not perform spatial averaging is much higher.

Answer: We have changed it to "partly" as suggested by the Reviewer.

13.Page 5, line 21: Remove the reference to Goddard et al (2013) here. They were not the ones defining the MSESS.

Answer: We have replaced "Goddard et al." by "Murphy, 1988" in the revised version.

14. Page 6, line 1: Two issues regarding the MSESS-formula:

14.1. Please replace the vertical bars (indicating the calculation of an absolute value) by brackets. Essentially it doesn't make a difference here but, you should follow the derivation presented by Murphy (1988).

Answer: We have replaced the vertical bars by brackets.

14.2. More generally; i wonder if it is useful to present a formula for the MSESS that holds only for the climatology as reference forecast. You present results, too, that are based on other reference forecasts, so it may be better to stay with the basic definition of MSE and MSESS as provided on page 5 already. If you want to present some decomposition, I suggest to stay with the more general one, I wrote down in my previous review. And given that these derivations are provided by other papers already, I don't see the need to indicate some derivation of the conditional bias in your paper from my point of view, you could directly provide the formula for the conditional bias as it is (see below). But please spend a few sentences, describing explicitly that the MSESS (partly) depends on the correlation but also on the condictional bias. This would be an important step, helping the reader to establish links between your MSESS- and ACC-results. This is what I asked for in my previous review.

Answer: We followed this suggestions an added the more general form of the MSESS decomposition in chapter 3 and extended the description of the dependency of the MSESS on the correlation and the conditional bias.

15. Page 6, line 4: Now the vertical bars are definitely wrong. Your tables contain negative values for the CB, too. So, it should be brackets (or nothing).

Answer: We have replaced the vertical bars by brackets as suggested.

16. Page 6, line 14: The correlation (ACC) is not only independent from the mean bias but also from the variance of the specific target variables.

Answer: We thank the Reviewer for this hint. We have added this information in the revised version.

17. Page 7, line 1-3: Please avoid using "bias" in this context. This might be misleading. Maybe just replace by "deviation", or something similar.

Answer: We have replaced it by "deviations" as suggested.

18. Page 7, line 19: Delete "of the hindcasts and decadal predictions".

Answer: We have deleted it in the revised manuscript.

19. Page 7, line 20: Replace "more reliable" by something like "better" or similar. Reliability has a specific meaning in forecast verification that is not meant here.

Answer: We agree that "reliable" is misleading here. We have now replaced it by "better predictions".

20. Page 8, lines 30-33: Two issues regarding your thersholds for coloring MSESS- and CB-values in the tabel:

20.1. Your thersholds in the text partly don't match the thresholds mentioned in the caption. Please check carefully!

Answer: We clarified the captions for Tables 1-3 and changed the colouring slightly to provide a clear and consistent description (see also answers to 20.2 and 21).

20.2. The choice of these thresholds seems totally arbitrary here. Is there some though behind it? If so, please explain. At least for the MSESS it would have made much more sense to me to also use a certain significance threshold as justification for coloring here.

Answer: The choice of the thresholds in Tables 1-3 is based on the typical level, above which the skill scores are regarded significant by the bootstrapping. This is about 0.3 for the MSESS

and about 0.4 for the correlation (highlighted in green). For the conditional bias it was in a range between +/-0.2. Negative MSESS and correlation values are highlighted in red, as well as CB values beyond +/-0.3. Values in between are regarded as not significant and therefore not marked. We have clarified this in the revised manuscript.

21. Page 9, line 1-3: Why do you suddenly use a t-test for assessing statistical significance? It would have been possible to use the same bootstrapping approach for CB_AV, too.

Answer: We agree that this was confusing. The conditional bias has an optimal value of zero. Therefore, the 0-hypothesys is different. To be consistent with the other scores, we now apply a common threshold of 5% (+/-0.05) to indicate a distinct difference of the skill scores between global and regional ensemble (added value), as well based on the typical level above which the differences are significant. Differences above 0.05 are now marked in green, below -0.05 in red and between +/-0.05 in white. We have changed the text and the captions accordingly.

22. Page 10, line 4-5: As mentioned above, it is no open question if there is a dependency. This would be one of the instances where I would ask you for a reformulation in the sense of "demonstrating skill bias dependency" (see my general comment 2 above).

Answer: We have rephrased this sentence and the remainder of this section according to comment 2.

23. Page 10, lines 8 and follwing, as well as Fig. 5: I think it might be quite misleading to demonstrate this bias dependency compared to the unitialzed simulations. They suffer from a bias, too, and currently it is not clear from the manuscript whether you reduce the number of unitialised simulations for the reference forecast, too, or use their full ensemble in every instance. I strongly suggest to demontrate the skill bias comparing to the climatology as reference forecast. This one is unbiased!

Answer: Originally we have exclusively focused on the uninitialized historicals as reference dataset, as is mostly done in the decadal forecast research community (see also references in our manuscript). This is motivated as follows: for future projections of the upcoming decades the RCP scenarios are used. When analysing past decades, the analogue to these RCP scenarios are the uninitialized historicals. Hence, a decadal prediction system is regarded as skilful, when the initialised hindcasts are closer to the observations than the uninitialized historicals, and these historicals are therefore used as reference. However, in one of the former revisions of our manuscript we additionally included the climatology as reference as suggested by one of the Reviewers, since we agreed with this Reviewer that this may improve the scientific value of our study. Nevertheless, our main focus still lies on the uninitialized historicals. And since we always use the same full ensemble of 10 members for the uninitialized historicals in Figure 5, we think that it is suitable for our purposes to demonstrate the improvement of the prediction skill when the number of ensemble members is increased. We have added this information in section 4.3. Therefore, depending on the Editors decision we would prefer to keep Figure 5 as it is in the revised manuscript.

24. Page 11, lines 3-7: I think you definitely should remind the reader here once again that your box-whisker plots do not contain the full uncertainty of the skill score estimates!

Answer: We are not sure if we fully got the intention of the remark, beyond what was already changed w.r.t to remark 23. Nevertheless, we added a remark in the caption of figure 5 to indicate that it just covers the uncertainty of the skill estimates due to the sample size.

25. Page 11, lines 11-12: This is also included in the CMIP6-DCPP requirements/recommendations,sp please include a reference to Boer et al. (2016) here, too.

Answer: We have included the reference to the DCPP and to Boer et al. (2016) in the revised manuscript.

26. Page 11, line 14: Please rephrase the research question and your following text in the sense of my general comment 3, so make it mor clear that you are addressing the issue of few initializations here.

Answer: We have rephrased the research question as suggested by the Reviewer and reformulated the following text accordingly.

27. Page 11, line 23: Replace "starting years" by "initializations only".

Answer: We have changed it to "initializations" in the revised manuscript.

28. Page 11, line 24: Replace "starting years" by "initializations".

Answer: We have changed it to "initializations" in the revised manuscript.

29. Page 12, line 5: Replace "predictability in" by "prediction skill of".

Answer: We have rephrased it as suggested.

30. Page 12, line 27-28: Rephrase. Maybe something like "Based on MPI_b1 data [...] we could show that results derived from only those five initializations used in our study qualitatively agree with results based a full set of annual initializations".

Answer: We have rephrased it to "Based on the MPI_b1 data, it was shown that results derived from only five initializations used in this study qualitatively agree with results based on the full set of annual initializations."

31. Page 12, line 32: Replace "predictability" by "prediction skill".

Answer: We have changed it as suggested.

32. Page 12, line 19: Be cautious with such statements regarding the relevance of drifts (and subsequent corrections) in case of anomaly initializations. Some studies show that these feature drifts, too. Reformulate to something like "the general expectation is that drift correction is less important for prediction systems employing anomaly initialization".

Answer: We thank the Reviewer for this hint. We have changed it accordingly.

33. Page 12, line 22: Replace "starting dates" by "initializations".

Answer: We have changed it to "initializations".

34. Page 14, line 1: Replace "predictability" by "prediction skill".

Answer: We have changed it as suggested.

35. Page 14, line 3: Maybe it's worth mentioning that this huge amount of 1000 RCM model years may also be a valuable set for other studies, not necessarily related to decadal prediction. I mean, it's a huge dataset in comparably high resolution representing the European climate of the recent past…

Answer: We agree that such a sample is very valuable not only for decadal predictions but also beyond. We have added this information in the revised version and named return periods of extreme events as example.

---

## Author Response (AR4)

**Reply to Editor's comments for manuscript esd-2017-70**

Comment by the authors:

Dear Editor,

5 We are grateful for your decision on our manuscript. We have carefully corrected the manuscript according to your comments. We will submit, together with all mandatory files, a manuscript version in the File Manager with changes according to your comments marked in red (given below).

Many thanks and best regards,

10 Mark Reyers (on behalf of the co-authors)

[revised manuscript text omitted]